# Microscopic mechanisms of pressure-induced amorphous-amorphous transitions and crystallisation in silicon

Zhao Fan [1] & Hajime Tanaka [1,2] ✉

Some low-coordination materials, including water, silica, and silicon, exhibit polyamorphism, having multiple amorphous forms. However, the microscopic mechanism and kinetic pathway of amorphous-amorphous transition (AAT) remain largely unknown. Here, we use a state-of-the-art machine-learning potential and local structural analysis to investigate the microscopic kinetics of AAT in silicon after a rapid pressure change. We find that the transition from low-density-amorphous (LDA) to high-density-amorphous (HDA) occurs through nucleation and growth, resulting in non-spherical interfaces that underscore the mechanical nature of AAT. In contrast, the reverse transition occurs through spinodal decomposition. Further pressurisation transforms LDA into very-high-density amorphous (VHDA), with HDA serving as an intermediate state. Notably, the final amorphous states are inherently unstable, transitioning into crystals. Our findings demonstrate that AAT and crystallisation are driven by joint thermodynamic and mechanical instabilities, assisted by preordering, occurring without diffusion. This unique mechanical and diffusion-less nature distinguishes AAT from liquid-liquid transitions.

It is well-known that some materials have two or more crystalline forms at a fixed composition, e.g., graphite and diamond for carbon, known as polymorphism[1]. Analogously, some systems have two or more amorphous forms, known as polyamorphism, and the transition between the different amorphous states is called amorphous-amorphous transition (AAT)[2–8]. AAT has been observed experimentally upon pressurising various materials, including water[9], silicon[10,11], oxide glasses[12], chalcogenide glasses[13], metallic glasses[14], and phase-change materials[15], all of which are close-relevant to our daily life and living environment and broadly applied in industry. Therefore, a deep understanding of AAT is essential to predict and create new materials for technological advances. In addition, it is of fundamental importance to understand the relationship between the solid-state AAT and the liquid-liquid transition (LLT), as the two phenomena are expected to be closely related to one another[2–8,16–19].

The nature of AAT remains elusive due to its strongly out-of-equilibrium nature[2–8,16–19]. There is still considerable debate regarding whether AAT is a continuous process or indeed possesses a truly first-order nature, resembling phase transitions between thermodynamic equilibrium states such as crystals and liquids. For silica, for example, Lacks argued in 2000 that the AAT in silica should be a first-order transition based on classical molecular simulations results[20]. However, Hasmy et al. recently suggested that the structural changes from low- to high-density amorphous structures in silica proceed through a sequence of percolation transitions according to their ab initio molecular dynamics (MD) simulations[21]. This controversy is partly due to the lack of a well-defined local structural order parameter[22] that can describe AAT. Moreover, specific atomic-level evidence is still missing as to whether AAT can take place via nucleation and growth (NG) or spinodal decomposition (SD), similar to thermodynamic phase transitions[23]. In addition, over the years, more and more evidence indicates that the transformation into a new product phase from a parent phase occurs through an intermediate state in

[1]Research Center for Advanced Science and Technology, The University of Tokyo, 4-6-1 Komaba, Meguro-ku, Tokyo 153-8904, Japan. [2]Department of Fundamental Engineering, Institute of Industrial Science, University of Tokyo, 4-6-1 Komaba, Meguro-ku, Tokyo 153-8505, Japan. ✉e-mail: tanaka@iis.u-tokyo.ac.jp

crystallisation[24,25], e.g., ice nucleation[26], the nucleation of diamond crystals[27], and the solid-state phase transition in metallurgic systems[28]. So far, it remains unclear whether analogical non-classical transition pathways are involved in AATs.

MD simulation is a promising and powerful tool to resolve these issues regarding AAT, as it can provide atomic-level details of the transition processes. For silicon, the classical Stillinger-Weber (SW) model[29] has been known to be a computationally efficient model. The existence of an LLT in the SW silicon has been confirmed through many years of efforts[30–32]. However, this model is not suitable for studying AATs properly[33] because it is difficult to reproduce the behaviour of silicon in a very high-pressure region. For example, the simple hexagonal (sh) crystal, which is hypothesised to be the crystalline counterpart of the very-high-density amorphous (VHDA) form of silicon[34], has been shown an unfavourable phase in the SW potential[35]. Recently, Deringer et al.[33] demonstrated that a machine learning (ML) potential[36] is effective in simulating AAT in silicon under pressure. They observed that low-density amorphous (LDA) and high-density amorphous (HDA) regions coexist, rather than appearing sequentially, before collapsing into an intermediate VHDA form and finally crystallising into the sh phase when increasing hydrostatic pressure $P$ on the LDA form of amorphous silicon (a-Si) at 500 K from 0 to 20 GPa with a constant rate of 0.1 GPa ps$^{-1}$. This work showed the power of the ML potential in studying AAT; however, continuous pressurisation employed to induce AAT is not suitable for elucidating the kinetic pathway and mechanism of AAT. Therefore, the fundamental questions mentioned above regarding AAT in silicon are yet to be answered.

One potential explanation for why Deringer et al. did not observe a complete transition between LDA and HDA[33] could be attributed to the continuous pressurisation, sustained at a rate of 0.1 GPa ps$^{-1}$, which was utilised in their simulations. In contrast, our approach involves a different pressure elevation protocol that is better suited for investigating the kinetics of phase transformations. To be more precise, we executed a rapid linear pressure increase on the a-Si sample, raising it to a desired level within the range of 10–15 GPa at 300 K within 100 ps. This can be approximated as an almost instantaneous alteration in pressure. Subsequently, we followed the relaxation process for up to 3 ns under isothermal-isobaric conditions.

Here, we utilise advanced machine-learning potential and local structural analysis to explore the microscopic kinetics of AAT in silicon following a rapid pressure change. We identify three amorphous forms at ambient temperature: LDA, HDA, and VHDA. Emphasising differences in short-range structure, particularly the underlying structural order parameters governing AAT, we reveal kinetic pathways and mechanisms of transitions between these forms and the crystallisation of denser amorphous states. Specifically, we observe the LDA-to-HDA transition with non-spherical interfaces via nucleation and growth, exposing the mechanical nature of AAT. Conversely, the reverse transition undergoes spinodal decomposition. Further pressurisation transforms LDA into VHDA, using HDA as an intermediate state. Finally formed VHDA states are inherently unstable, transitioning into crystals. Our findings underscore joint thermodynamic and mechanical instabilities driving AAT and crystallisation, assisted by preordering without diffusion. Importantly, our research illuminates the crucial roles of preordering in AATs and crystallisation, further providing insights into mechanistic contributions arising from diffusionless solid-state transformations, such as the mechanical collapse of structural motifs with a significant local volume reduction.

## Results and discussion
We generated an as-quenched a-Si model containing 8192 atoms at zero pressure using a recently developed ML potential[36] used in ref. 33. We rapidly increase pressure $P$ of the as-quenched a-Si from 0 GPa to various target pressures (a linear increase of $P$ within 100 ps) and then follow isothermal-isobaric relaxation for a long time at ambient temperature ($T = 300$ K), which is similar to usual experimental protocols. We use coarse-grained local bond orientational order parameters[37,38] and a convolutional neural network (CNN) model[39,40] to identify different local crystalline and amorphous atomic environments (see Methods for more details about MD simulations and identifying different local atomic environments).

We emphasise that this protocol of the isothermal-isobaric relaxation after a rapid pressure change is essential for studying the phase-transition kinetics. We note that the AATs are strongly dependent on the pressurisation rate, and we did not observe the LDA-HDA transition at either 300 or 500 K when increasing $P$ continuously from 0 to 20 GPa at a constant rate, even with a pressurisation rate of 0.01 GPa ps$^{-1}$ (see Supplementary Note 1). We confirmed that our sample size is large enough for studying the AATs (see Supplementary Note 2).

### Structural characteristics of three amorphous forms and two crystals
Typical transition behaviours observed at different conditions are showcased in Fig. 1a, Supplementary Figs. 1 and 2a–c (see also below and Supplementary Note 3 for further details). In this section, we will highlight the main structural characteristics of the three amorphous forms of a-Si. More structural characterisations are described in Supplementary Note 4.

First, we focus on the structure of low-pressure a-Si, LDA, and its stability in a low-pressure range. We chose a state after relaxing an as-quenched a-Si sample at 10 GPa for 200 ps (denoted as LDA$_{10,200}$ in Fig. 1a) as a typical LDA state and analysed its structure. We observed a wide gap with almost zero intensity between the first and second peaks of its $g(r)$ (see Fig. 1b), a single prominent peak around 109° (the characteristic angle of a tetrahedron) in its bond angle distribution function (BADF) (Fig. 1c), and more than 95.8% of atoms with a coordination number (CN) of 4 (the average CN of 4.04) (Fig. 1d). These observations indicate that most atoms in this a-Si form are centred on distorted tetrahedra. The minor peak around 55° in its BADF should result from a small fraction of atoms with CN = 5. All these features are consistent with previous ab initio MD simulation results of the LDA form of silicon[34], suggesting the as-quenched a-Si to be LDA. It is stable, at least within the timescale accessible to the current MD simulation, when increasing $P$ up to 11 GPa (see Supplementary Fig. 3).

Next, we consider the structural characteristics of HDA. We take the sample after relaxing 1 ns at 12 GPa as a typical HDA state. Its $g(r)$ is distinctly different from that of LDA but without long-range order (see Fig. 1b). The most pronounced differences are: (i) the position of the first peak moves to the right side of the first peak of LDA, although its volume is lower than that of LDA; (ii) the intensity of the trough between the first and second peaks in $g(r)$ is now much larger than zero; (iii) the CN of the HDA form is dominantly CN = 6 (the average CN = 6.09) (see Fig. 1d), and the highest peak on its BADF is located around 90° (see Fig. 1c). These features are consistent with those of HDA of a-Si reported in ref. 34, suggesting the amorphous form obtained at 12 GPa to be HDA.

As seen from Fig. 1a, the magnitude of the volume drop is much more significant when increasing $P$ from 0 to 15 GPa within 100 ps than when increasing $P$ from 0 to 12 GPa. This suggests a structural transformation occurring during a rapid pressure increase to 15 GPa, in addition to elastic shrinkage. The structural transition finishes after relaxing the sample at 15 GPa for a short time (~10 ps). This amorphous form appearing at 15 GPa is unstable and quickly crystallises during the subsequent relaxation. The CN of the amorphous form obtained at 15 GPa is dominantly CN = 8 (the average CN = 8.21) (see Fig. 1d), and the highest peak on its BADF is located around 60° (see Fig. 1c). These features are consistent with those of VHDA[41], indicating this amorphous form to be VHDA.

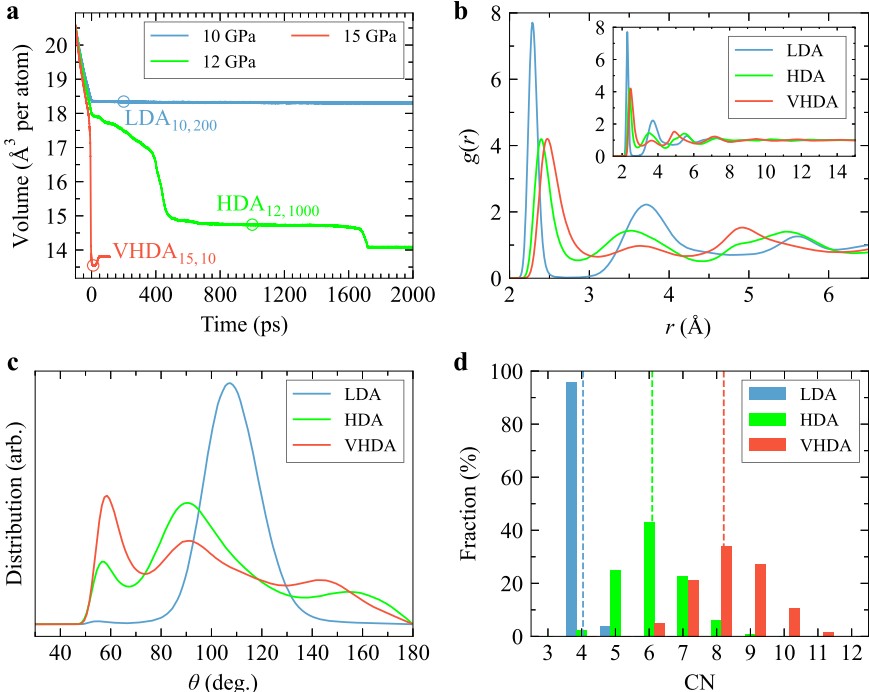

**Fig. 1 | Three amorphous forms of silicon and their structural differences. a** The variation of average atomic volume with time at 300 K when relaxing the as-quenched a-Si at three different constant pressures. During the first 100 ps (from −100 to 0 ps), the pressure was ramped linearly to the various target values from 0 GPa. **b**–**d** show the radial distribution function $g(r)$, bond angle distribution function (BADF), and distribution of coordination number (CN), respectively, for the three amorphous states of silicon indicated by circles in **a**, i.e., low-density amorphous (LDA), high-density amorphous (HDA) and very-high-density amorphous (VHDA). The inset of **b** shows $g(r)$ over a longer radial range up to 15 Å to confirm there is no long-range order in the three amorphous forms. The vertical dashed lines in **d** represent the average coordination number (CN) of the three forms. Here we used a cutoff of 2.85 Å, the same as in ref. 33, to determine the first neighbouring shell. Source data are provided as a Source Data file.

The density and CN distribution are the most pronounced differences among the three amorphous forms. Furthermore, a single coarse-grained local bond orientational order parameter $\bar{q}_4^{21}$ (see Methods for the definition) can identify local LDA-like environments from the other two local amorphous environments (Supplementary Fig. 4d). In addition, a CNN model[39,40] can separate HDA- and VHDA-like environments with an accuracy of >99.66% (Supplementary Fig. 4f, see Methods for the details).

We also characterise the crystals that form after AAT under pressurisation. Coarse-grained local bond orientational order parameters $\bar{q}_4^8$ and $\bar{q}_6^8$ can be used to identify local $\beta$-Sn- and sh-like environments from local amorphous environments (Supplementary Fig. 4a–c). Supplementary Fig. 5 shows atoms centred on different local environments with different colours in a typical configuration for each of the three amorphous forms. Using these local structural order parameters, we also found that the crystallisation product at 12 GPa is dominated by $\beta$-Sn-like environments (74.1%), in addition to 25.6% of sh-like environments and a small fraction of residual amorphous environments. On the other hand, the crystallisation product at 15 GPa is dominated by sh-like environments (82.0%), in addition to 16.3% of $\beta$-Sn-like environments and a small fraction of residual amorphous environments. The results are also supported by the $g(r)$ similarity of the crystallisation products with bulk $\beta$-Sn and sh crystals (see Supplementary Fig. 6a). They are also consistent with Morishita's hypothesis[34] that the crystalline counterparts of LDA, HDA, and VHDA forms of a-Si should be diamond, $\beta$-Sn, and sh structures, respectively, according to their resemblance in CN. Supplementary Fig. 6b, c show structural snapshots of the crystallisation products at 12 and 15 GPa, respectively.

For describing thermodynamic LLT theoretically, a multiple order parameter model was introduced[22,42], based on the concept of locally favoured structures with a specific orientational symmetry, whose fraction is treated as an additional rotationally invariant scalar order parameter besides density that describes a vapour-liquid transition. We now identify structural order parameters relevant to AAT from the above structural characterisations. The locally favoured structure in the LDA form of silicon should be a regular four-folded polyhedron—tetrahedron[43,44], which is coincidently the local polyhedron in its crystalline counterpart—diamond structure. Given that the dominant CN in the HDA form is 6, and the highest peak of the BADF of the HDA form is located around 90° (see Fig. 1c, d). This suggests that the locally favoured structure of HDA has octahedral symmetry. Supplementary Fig. 7 illustrates characteristic octahedral environments displaying different levels of distortion surrounding atoms that exhibit a CN of 6 in the HDA form. The small peak observed around 60° in the BADF (Fig. 1c) can be attributed to atoms with CN ≠ 6, and it signifies distortions occurring within octahedral structures. These distortions may be closely linked to the structural instability that leads to higher-coordination structures or configurations resembling VHDA structures (Supplementary Fig. 8b). We made further structural analysis focusing on the similarity of the local symmetry of HDA to those of a regular octahedron and $\beta$-Sn. The result shows that they are very similar, but more weight on $\beta$-Sn (see Supplementary Fig. 9b and Supplementary Note 5). However, this result could be attributed to the inherent distortion of octahedral symmetry. Thus, we infer that the locally favoured structure of HDA has octahedral orientational symmetry, which needs to be checked in the future.

We also tried to identify the locally favoured structure in the VHDA form but found that even the dominant eight-folded polyhedra in the VHDA form are pretty irregular, and it is hard to extract some common features, except that these eight-folded polyhedra favour bond angles around 60° (see Fig. 1c). The high degree of irregularity of eight-folded polyhedra in the VHDA form is also reflected in their significant deviation from the perfect local sh polyhedron (note that the sh crystal

is the crystalline counterpart of the VHDA form, see ref. 34) (see Supplementary Fig. 9c). We also found that the local atomic environments centred on atoms with the same CN are comparable between HDA and VHDA (see Supplementary Note 6). This suggests that the primary difference within the first neighbouring shell between HDA and VHDA lies in the fraction of atoms with different CN. This observation supports a multiple-order-parameter model of LLT and AAT[8,22,42]. It is worth noting that, according to this model, two structural order parameters, in addition to the density order parameter describing the gas-liquid transition, are necessary to account for the existence of three distinct amorphous states[8,22]. Additionally, apart from the pronounced difference in CN, there are structural distinctions within the intermediate range between HDA and VHDA forms. This is supported by the radial distribution function, $g(r)$ (depicted in Fig. 1b), as well as the ring analysis (Supplementary Fig. 10). Furthermore, there are interesting distinctions among the three amorphous states in terms of the density difference between an amorphous state and its corresponding crystalline counterpart and the shift in the first peak of $g(r)$ with increasing pressure, see Supplementary Fig. 11 and Supplementary Note 4.

Moreover, we calculated the vibrational density of states (VDOS) for each amorphous form using the Fourier transformations of the velocity auto-correlation functions (VACFs)[45,46]. The VDOSs, denoted as $g(\omega)$, as well as those normalised by $\omega^2$, represented as $g(\omega)/\omega^2$, are shown in Supplementary Fig. 12. It is evident that three amorphous states exhibit distinct VDOSs. In the case of VHDA, there is a prominent quasi-elastic component in $g(\omega)/\omega^2$, indicating its instability. This feature of VHDA aligns with its disordered local structures, which make it susceptible to crystallisation. Even in the case of HDA, a similar quasi-elastic component in $g(\omega)/\omega^2$ is observed shortly after the transformation from LDA, but it gradually diminishes as mechanical stability is acquired over time (see Supplementary Fig. 13). In the future, it will be essential to conduct a comprehensive study on the thermodynamic and mechanical stability of these high-pressure amorphous forms, especially VHDA.

According to Supplementary Fig. 9, the degree of deviation between the dominant local polyhedra of the three amorphous forms and its perfect local polyhedron in the corresponding crystalline counterpart is the smallest for the LDA form and most significant for the VHDA form. Naturally, a pressure increase enhances structural disorder in a dense solid state, where structural relaxation is limited. The more substantial structural disorder not only increases the thermodynamic driving force for crystallisation but also makes a system less resistive against pressure-induced collapse. This may explain why the resistance to crystallisation is the highest for the LDA form and lowest for the VHDA form among the three amorphous forms.

## LDA-HDA transition

Now, we turn our attention to transition kinetics. During the linear ramp of $P$ from 0 to 12 GPa, the increase in the fraction of HDA-like atoms is negligible (Supplementary Fig. 14a); thus, the decrease in the sample volume should be mainly due to elastic shrinkage. Then, in the subsequent isothermal-isobaric relaxation, LDA transforms into HDA, during which the sample volume drops by 17.0% (from 17.99 Å$^3$ to 14.93 Å$^3$) over the first 500 ps (see Fig. 1a). The HDA subsequently crystallises into β-Sn crystals, resulting in a quick sample volume drop by 2.4% around 1.7 ns (see Fig. 1a). The crystallisation occurs via NG, as demonstrated in Supplementary Fig. 15.

We also conducted a structural analysis of the LDA-HDA transition to determine whether the AAT occurs via NG or SD[8,22,47]. As seen from Fig. 2a, b, the size of the largest HDA cluster fluctuates within a range between 6 and 30 atoms during the first 50 ps in the isothermal-isobaric relaxation at 12 GPa, then jumps to ~110 atoms over the short period from 50 to 65 ps, and continuously increases afterwards. These

results indicate that the LDA-HDA transition proceeds via NG. The first 50 ps is the incubation period before nucleation; thus, the critical nucleus size is between 110 and 240 atoms. In our cluster analysis, we employed a cutoff distance of 2.85 Å, consistent with the approach used in ref. 33. Importantly, we verified the robustness of our cluster analysis results by examining their independence from the chosen cutoff value within a range extending from 2.85 Å (corresponding to the first trough of the radial distribution function, $g(r)$) to 3.72 Å (corresponding to the second peak of $g(r)$ for LDA), as demonstrated in Supplementary Fig. 16. Furthermore, Fig. 2c, d, show the evolution of the CN distribution and atomic volume distribution, respectively (see Supplementary Fig. 14c–e, respectively, for the evolution of $g(r)$, BADF, and distribution of the tetrahedral order parameter[48,49]). The variation of the atomic volume distribution during the LDA-HDA transition (Fig. 2d) is very similar to that during the crystallisation process shown in Supplementary Fig. 15d, further corroborating that the LDA-HDA transition takes place through NG. However, it should be noted that even for a thermodynamic transition, a clear distinction between NG and SD is valid only in the mean-field limit. In systems with short-range interactions near the boundary between the metastable and unstable regions, the transition exhibits a mixed nature[50].

We find that the structural changes occurring during the LDA-HDA transition can be effectively characterised by a systematic temporal evolution of the distribution of the local bond orientational order parameter $\bar{q}_4^{21}$. Specifically, a new peak emerges around $\bar{q}_4^{21} = 0.02$ and steadily increases while maintaining its position (as shown in Fig. 2e). This behaviour is reminiscent of the order parameter evolution observed in the thermodynamic NG process[8,22,47]. Therefore, $\bar{q}_4^{21}$ can serve as a suitable local structural order parameter[8,22] to effectively describe the LDA-HDA transition. This discovery of the NG-like features in AAT further supports the notion that AAT shares similarities with genuine thermodynamic phase transitions, which has been recognised as an intriguing and fundamental question in materials science[21]. However, it is important to emphasise that the AAT process in a solid state also involves mechanical contributions, as we will demonstrate later on.

Here it is worth noting that when calculating the local bond orientational order parameter $\bar{q}_l^N$, the choice of the parameter $l$ is not the only factor influencing its behaviour. The number of neighbouring atoms $N$ also plays a pivotal role in determining the distribution of $\bar{q}_l^N$ across various phases. For example, the distributions of $\bar{q}_4^8$ and $\bar{q}_4^{21}$ for the three amorphous forms of silicon exhibit distinct differences; see Supplementary Fig. 4a, d. This aspect can potentially limit the effectiveness of $\bar{q}_l^N$ in distinguishing between different phases, particularly when relying solely on a single method for determining neighbour lists.

A recent research[51] has pointed out that $\bar{q}_l$ may not be consistently effective in distinguishing between an amorphous solid and a high-density liquid in a core-softened model. We hypothesise that the reason behind this limitation might be attributed to the sole use of Voronoi tessellation in constructing neighbour lists. Indeed, $\bar{q}_l^N$ is not always effective in differentiating between various amorphous forms and/or liquid states. For instance, as detailed in Methods, identifying a single $\bar{q}_l^N$ that reliably distinguishes HDA from VHDA in silicon proves to be a challenging task (Supplementary Fig. 4e, h).

Recent studies, e.g., refs. 24,25,42,52, provided more and more evidence suggesting that preordering has a critical influence on crystal nucleation in supercooled liquids. Since the LDA-HDA transition takes place through NG, it is interesting to explore whether there is any preordering effect in the nucleation of HDA in LDA. We thus calculated the $\bar{q}_4^{21}$ distribution of the 151 atoms, which would constitute the HDA nucleus upon relaxation at 12 GPa for 100 ps, in the initial as-quenched sample at 0 GPa. The 151 atoms that would form the HDA nucleus upon pressurisation are indeed the atoms with relatively lower $\bar{q}_4^{21}$ in the initial sample (see Supplementary

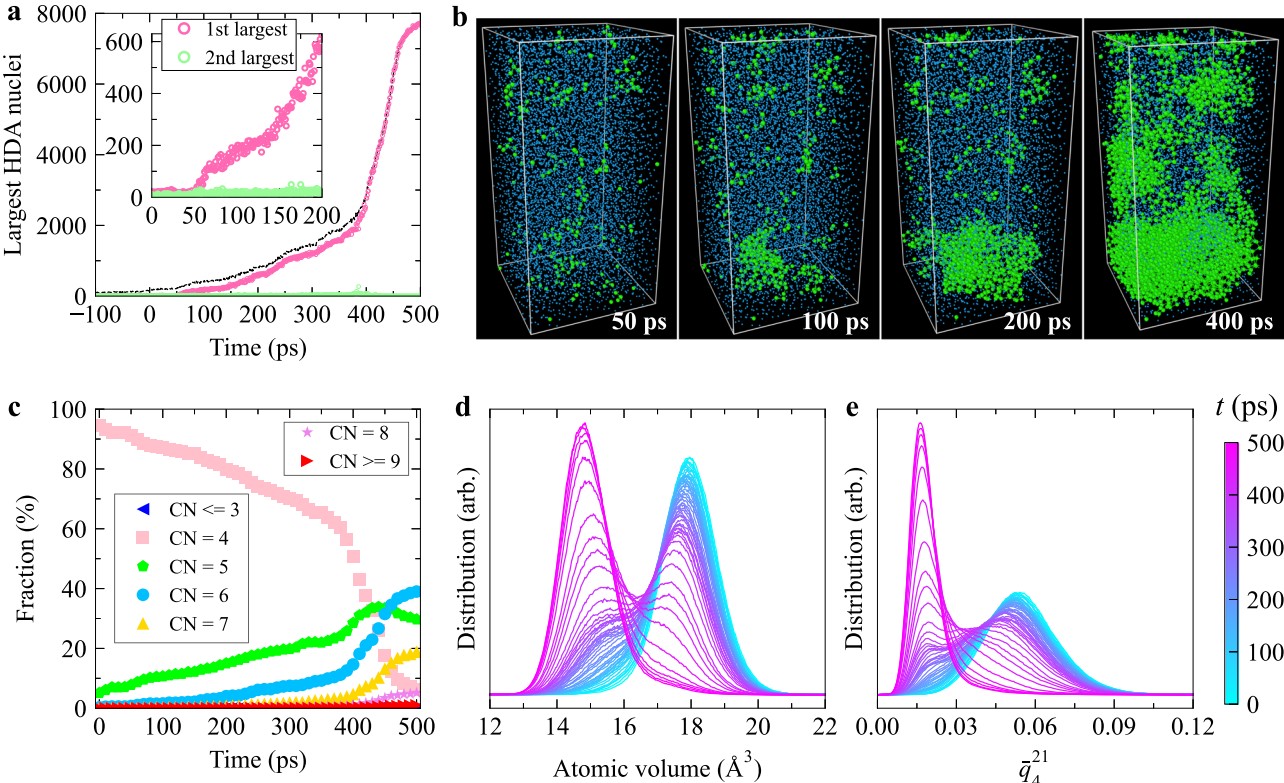

**Fig. 2 | The low-density amorphous (LDA) to high-density amorphous (HDA) transition behaviour.** This transition occurs when relaxing the as-quenched a-Si sample at 12 GPa. **a** The size evolution of the first several largest HDA nuclei. The dashed line represents the total number of HDA-like atoms. The inset highlights the initial stage of the transition. A cutoff of 2.85 Å was used when conducting cluster analysis. We verified that the results of the cluster analysis remain consistent across a range of cutoff values between 2.85 and 3.72 Å, see Supplementary Fig. 16.

**b** Typical structural snapshots during the LDA-HDA transition. Blue and green spheres represent LDA- and HDA-like atoms, respectively. There is no other type of local atomic environment in the configurations exhibited here. The atom size is adjusted for different structural environments for clarity. **c**–**e** show the variation of the coordination number (CN) distribution, atomic volume distribution, and distribution of coarse-grained local bond orientational order parameter $\bar{q}_4^{21}$, respectively, during this transition. Source data are provided as a Source Data file.

Fig. 14b). This implies that preordering (atoms with relatively lower $\bar{q}_4^{21}$) indeed assists HDA nucleation in LDA.

The significant role played by preordering in initiating AAT shares similarities with the influence of crystal-like preordering in crystal nucleation[24,25,42]. This underscores the general significance of preordering in first-order phase transitions. While thermodynamic ordering from a liquid is often driven by preordering, with a primary thermodynamic impact, the situation can be substantially different in pressure-induced solid-state phase transitions due to their mechanical nature. In this context, understanding how HDA-like preordering catalyses the nucleation of HDA domains, both thermodynamically and mechanically, presents an intriguing question. In addition to local symmetry matching, which reduces the energy barrier associated with the formation of new interfaces, the lower mechanical stability of preordered regions plays a crucial role in reducing the mechanical work involved in the transformation. Given the enhanced stability of HDA-like order under high pressure, it is intuitively expected that the transformation of HDA-like preordering into HDA domains becomes more feasible under compression. The intricate interplay between thermodynamic and mechanical factors within this transformation poses an intriguing question. However, due to the complexity of disentangling these intricate contributions, we leave this issue for future investigations.

Regarding the influence of preordering on AAT, it is worth noting that the LDA-HDA transitions observed at both 12.5 and 13 GPa for the same initial state exhibit similarities to those previously discussed (please refer to Supplementary Fig. 17 and additional discussion in Supplementary Note 7). Interestingly, both the HDA nucleus

containing 95 atoms at 12.5 GPa (case 1) after relaxation for 20 ps and the nucleus with 151 atoms at 12 GPa after relaxation for 100 ps share a common set of 61 atoms that were part of the initial as-quenched sample. This commonality suggests that the initiation of HDA nucleation occurred in approximately the same location at both 12 and 12.5 GPa, providing further evidence for the substantial influence of preordering on HDA nucleation. In other words, this observation implies that nucleation sites in solid-state phase transformations are determined by the initial solid structure, as opposed to the stochastic nature of nucleation from a fluid.

To further validate this notion, we conducted two additional simulations, each involving an instantaneous jump in $P$ from 0 to 12 GPa on the same initial a-Si sample at 300 K. These simulations were preceded by the assignment of distinct initial velocity fields. Subsequently, an isothermal-isobaric relaxation spanning 300 ps was performed. Remarkably, the first HDA nuclei consisting of 151 atoms, observed across both cases, and even during relaxation following a linear pressure ramp from 0 to 12 GPa, commonly share 42 atoms, as depicted in Supplementary Fig. 18. This result strongly supports the conclusion that nucleation sites during AAT in silicon at room temperature are indeed determined by the initial structure. Furthermore, investigating whether the incubation period preceding nucleation in AAT adheres to an exponential distribution presents an intriguing avenue. However, addressing this question would require quenching multiple independent samples, which is computationally demanding. Therefore, we leave it for future studies.

Finally, we mention a unique feature of AAT absent in LLT[8,19]. As we can see from Fig. 2b, the HDA nucleus has an irregular shape,

unlike LLT, where a nucleus of the new liquid phase tends to become spherical via diffusion[53,54]. This behaviour indicates the mechanical nature of AAT proceeding without diffusion in a solid state while accompanying elastic deformation, similar to crystallisation in an amorphous solid state[55,56]. The non-spherical nature of an HDA nucleus becomes evident through the analysis of the surface area-to-volume ratio as it evolves (Supplementary Fig. 19a). Notably, there is no consistent trend of the nucleus developing a spherical shape over time. This observation emphasises the combined influence of both thermodynamics and mechanics during the growth process. It is important to note that the interface energy does not play a central role in this process, as the transformation is driven by non-diffusive mechanical collapse rather than thermal diffusion. This observation is further supported by the displacement fields in the nucleation-growth process of HDA (see Supplementary Fig. 19b). These displacement fields lack coherent motion aimed at minimising the interfacial area of HDA domains. These findings collectively highlight the distinctive characteristics of non-diffusive solid-state transformations[55,56]. In addition, we evaluated the atomic-level pressure changes during an LDA-HDA transition, as illustrated in Supplementary Fig. 19c. It is noteworthy that a subtle pressure contrast between the two phases is observable, along with a widespread atomic-level pressure dispersion even after undergoing a coarse-graining process. Moreover, we have observed a non-uniform spatial distribution of the pressure in its coarse-grained form, as highlighted in Supplementary Fig. 19d. These results together serve as distinctive indicators of the mechanical characteristics inherent in solid-state transformations, a trait that is absent in their counterparts in the liquid state.

## HDA-LDA transition

When relaxing the HDA$_{12,1000}$ sample at 1 GPa after quickly reducing pressure from 12 to 1 GPa at a rate of 10 GPa ps$^{-1}$, the fraction of LDA-like atoms can quickly increase from 4.4% to above 80.0% within 7 ps (see Supplementary Fig. 2b). We can see a large LDA cluster containing more than 100 atoms just when the pressure reaches 1 GPa, probably due to a residual LDA in HDA. Nevertheless, the spatial fluctuations of tetrahedral order emerge and grow before the rapid growth of the largest LDA cluster (see Fig. 3a, b). The evolution of the CN distribution, atomic volume distribution, and $\bar{q}_4^{21}$ distribution during the HDA-LDA transition are presented in Fig. 3c–e. These transformation behaviours indicate that the HDA-LDA transition occurs through SD. The HDA-LDA transitions induced by relaxation at 6 GPa and by a continuous pressure decrease also behave similarly to the above (see Supplementary Fig. 2d–h and Supplementary Note 8). The NG- and SD-type transformations for the forward and reverse LDA-HDA transitions, respectively, may reflect the difference in the mechanical barriers in addition to that in the thermodynamic one: the transformation of tetrahedra-dominant LDA to octahedra-dominant HDA may be mechanically more resistive than the reverse one. Finally, we note that the HDA-LDA transition was not observed during isothermal-isobaric relaxation when pressure is >6 GPa within the time scale accessible to the current simulations (see Supplementary Fig. 2c). The HDA-LDA transition may proceed via NG when pressure is >6 GPa if LDA is more stable. This hysteresis behaviour deserves further investigation.

## LDA-VHDA transition

When relaxing the LDA$_{10,200}$ sample at 15 GPa after quickly increasing pressure from 10 to 15 GPa at a rate of 10 GPa ps$^{-1}$, the LDA-VHDA

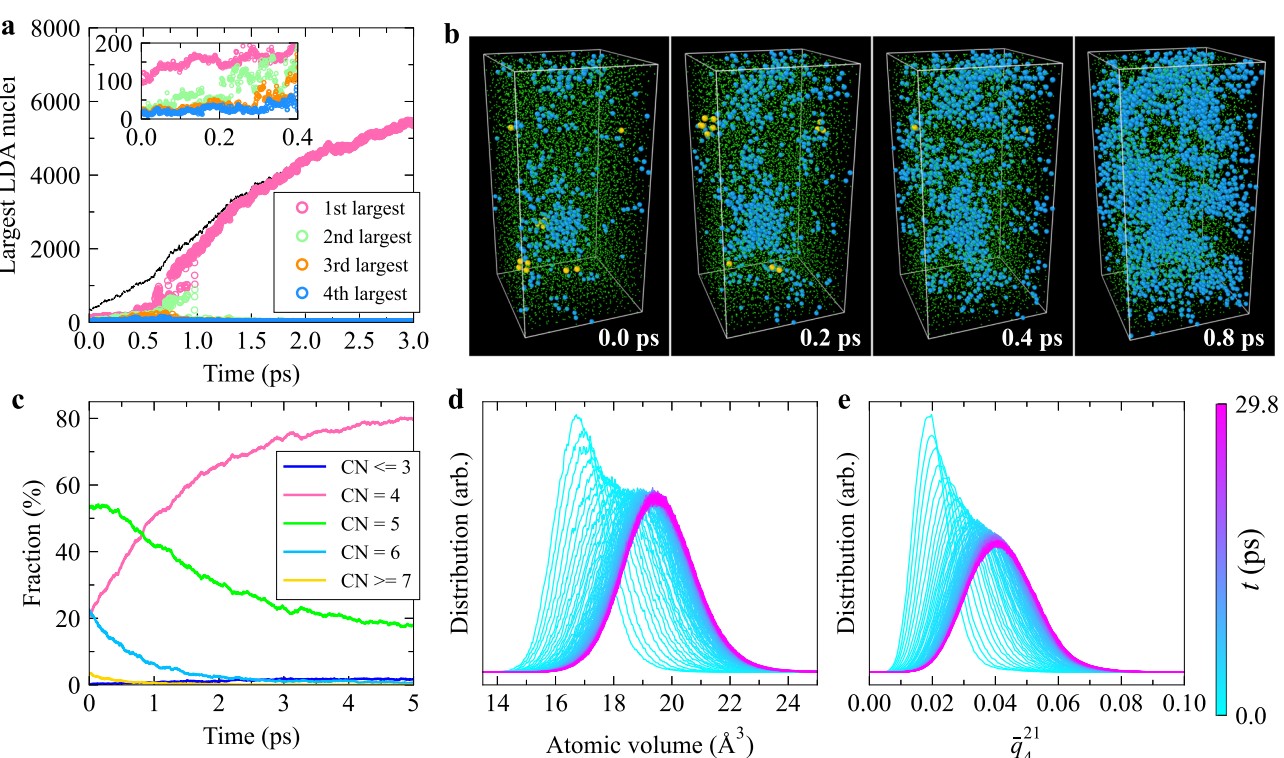

**Fig. 3 | The high-density amorphous (HDA) to low-density amorphous (LDA) transition behaviour.** This transition occurs when relaxing the HDA$_{12,1000}$ sample at 1 GPa. The pressure, $P$, on the HDA$_{12,1000}$ sample was first reduced quickly from 12 to 1 GPa at a constant rate of 10.0 GPa ps$^{-1}$ before relaxation. **a** The evolution of the size of the first several largest LDA nuclei. The dashed line represents the total number of LDA-like atoms. The inset highlights the initial stage of the transition. A cutoff of 2.85 Å was used when conducting cluster analysis. **b** Typical structural snapshots during the HDA-LDA transition. Blue, green, and yellow spheres represent LDA-, HDA-, and β-Sn-like atoms, respectively. There is neither very-high-density amorphous (VHDA)- nor simple hexagonal (sh)-like atom during the process considered here. The atom size is adjusted for different structural types for clarity. **c–e** show the variation of the coordination number (CN) distribution, atomic volume distribution, and distribution of coarse-grained local bond orientational order parameter $\bar{q}_4^{21}$, respectively, during the transition. Source data are provided as a Source Data file.

transition occurred through a two-step process, similar to the scenario known as Ostwald's step rule[57]. Specifically, during the LDA-VHDA transition, the fluctuations of octahedral order (or HDA clusters) grow very quickly (~2 ps) without an evident incubation period (see Fig. 4b and Supplementary Fig. 20a). Then, VHDA nuclei emerge exclusively inside the HDA regions (see the evidence presented in Fig. 4a, c) and then quickly grow such that 99.5% of atoms (8151 out of the 8192 atoms) in the supercell is VHDA-like after a short relaxation (9.9 ps) (see Supplementary Fig. 1c). Therefore, HDA is an intermediate state in the LDA-VHDA transition. This can be rationalised from the fact that the bond orientational order of the HDA form is intermediate between LDA and VHDA, i.e., both the $\bar{q}_6^8$ and $\bar{q}_{10}^{13}$ distributions of the HDA form are located in the middle between those of LDA and VHDA (see Supplementary Fig. 4b, e). The same applies to CN. This unique sequential AAT, i.e., the appearance of HDA clusters as an intermediate state in the LDA-VHDA transition, is selected since the LDA-HDA transition barrier is much lower than the direct LDA-VHDA one. Then, VHDA is preferentially formed in HDA regions due to their structural similarity and the resulting lower interfacial energy between them. The LDA-VHDA transitions occurring at other conditions also proceed through this two-step process (see Supplementary Fig. 20b, c and Supple-

mentary Note 9). As can be seen from Fig. 4a, the formation of VHDA in the LDA-VHDA transition appears to be an NG process. The rapid onset of the LDA-VHDA transition is catalysed by the intermediate HDA form.

## HDA-VHDA transition

When relaxing the HDA$_{12,1000}$ sample at 15 GPa, the highest fraction of atoms of VHDA-like local order reaches 80.8% at 30.8 ps, at which there are also 0.4% and 11.9% of LDA- and HDA-like atoms, respectively, and the largest crystal nucleus contains 418 ($\beta$-Sn- and sh-like) atoms (see Supplementary Fig. 1d (15 GPa, case 1)). These suggest that it is hard to obtain a homogeneous amorphous state of VHDA from HDA, unlike the LDA-VHDA transition, where the highest fraction of VHDA-like atoms can reach 98.9%. This is not surprising given that compared to LDA, HDA has more crystal precursors (see Supplementary Fig. 4a, b). We note that it is reasonable to assume that atoms with higher $\bar{q}_4^8$ or $\bar{q}_6^8$ act as crystal precursors (see the next section). According to Fig. 5a–c, the HDA-VHDA transition proceeds via the SD-like process, unlike the NG process of forming the VHDA form in the LDA-VHDA transition at the same pressure (see above). This difference is reasonable, considering that VHDA clusters only emerge from HDA but not LDA regions.

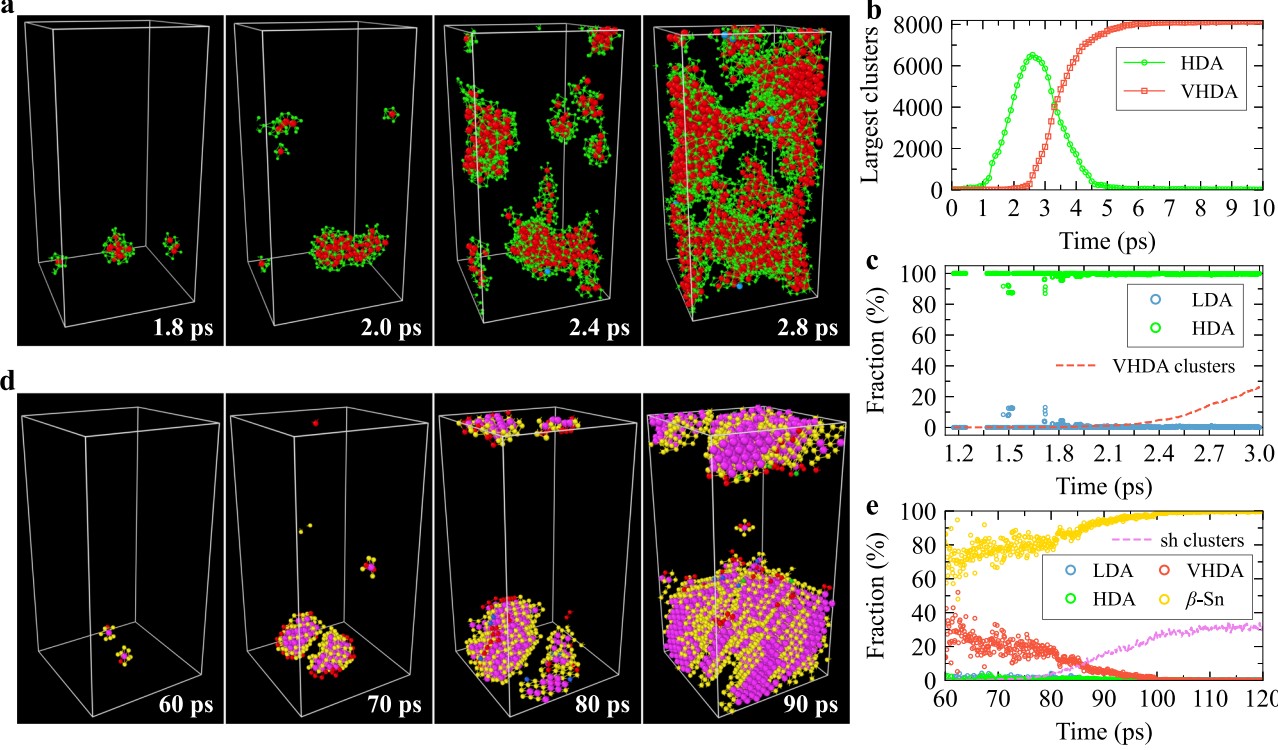

**Fig. 4 | The low-density amorphous (LDA) to very-high-density amorphous (VHDA) transition and subsequent crystallisation behaviours.** These transitions occur sequentially when relaxing the LDA$_{10,200}$ sample at 15 GPa (case 1). The pressure, $P$, on the LDA$_{10,200}$ sample was first increased quickly from 10 to 15 GPa at a constant rate of 10.0 GPa ps$^{-1}$ before relaxation. **a** Visualisation of the structural environment of the first neighbouring shell surrounding VHDA-like individual atoms and clusters during the emergence of VHDA-like atoms in the LDA-VHDA transition. Only VHDA-like atoms and their first neighbours are shown in each panel. Blue, green, and red spheres represent LDA-, high-density amorphous (HDA)-, and VHDA-like atoms. The atom size is adjusted for different structural types for clarity. There is no crystal-like atom over the time window considered here. **b** The size evolution of the largest HDA and VHDA clusters during the LDA-VHDA transition. **c** The structural environment of the first neighbouring shell surrounding VHDA-like individual atoms and clusters during the emergence of VHDA-like atoms in the LDA-VHDA transition. The circles in **c** with different colours (see legend) represent the fraction of atoms in different local structural environments within the

first neighbouring shell surrounding VHDA-like atoms and clusters. The red dashed line represents the fraction of VHDA-like atoms relative to all atoms in the entire supercell. There is no crystal-like atom during the first 3 ps of the relaxation. **d** Visualisation of the structural environment of the first neighbouring shell surrounding the clusters of simple hexagonal (sh) crystals in the subsequent crystallisation process. Only sh-like atoms and their first neighbours are shown in each panel. Blue, green, red, yellow, and magenta spheres represent LDA-, HDA-, VHDA-, $\beta$-Sn-, and sh-like atoms, respectively. The atom size is adjusted for different structural types for clarity. **e** The structural environment of the first neighbouring shell surrounding the clusters of sh crystals during the subsequent crystallisation process. The circles with different colours (see legend) represent the fraction of atoms in different local structural environments within the first neighbouring shell surrounding regions of sh crystal. The magenta dashed line represents the fraction of sh-like atoms relative to all atoms in the supercell. Source data are provided as a Source Data file.

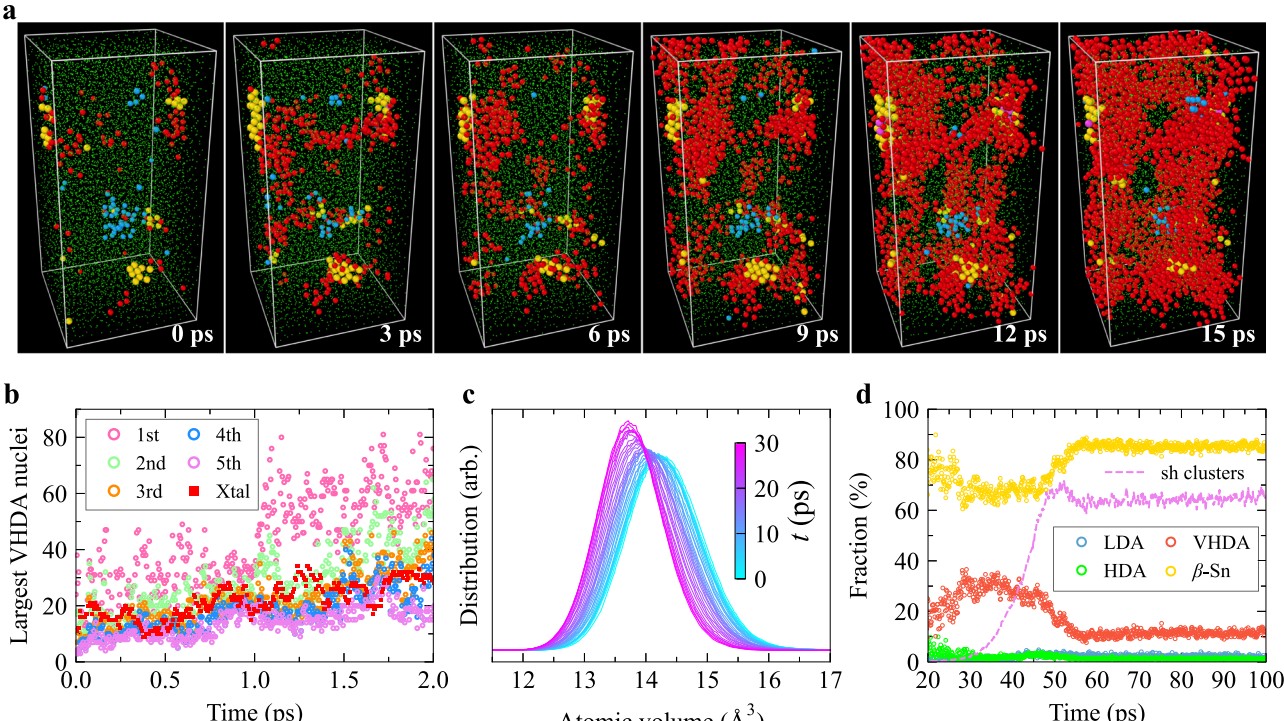

**Fig. 5 | The high-density amorphous (HDA) to very-high-density amorphous (VHDA) transition and subsequent crystallisation behaviours.** These transitions occur sequentially when relaxing the HDA$_{12,1000}$ sample at 15 GPa (case 1). The pressure, $P$, on the HDA$_{12,1000}$ sample was first increased quickly from 12 to 15 GPa at a constant rate of 10.0 GPa ps$^{-1}$ before the relaxation. **a** The structural snapshots during the HDA-VHDA transitions. Blue, green, red, yellow, and magenta spheres represent low-density amorphous (LDA)-, HDA-, VHDA-, $\beta$-Sn-, and simple hexagonal (sh)-like atoms, respectively. The atom size is adjusted for different structural types for clarity. **b** The evolution of the size of the first several largest VHDA nuclei during the HDA-VHDA transition. The solid red squares represent the size of the largest crystal nucleus (including both $\beta$-Sn-like and sh-like atoms). A cutoff of 2.85 Å was used when conducting cluster analysis. **c** The variation of atomic volume distribution during the HDA-VHDA transition. **d** The structural environment of the first neighbouring shell surrounding the clusters of sh crystals during the subsequent crystallisation process. The circles with different colours (see legend) represent the fraction of atoms in different local structural environments within the first neighbouring shell surrounding regions of sh crystal. The magenta dashed line represents the fraction of sh-like atoms relative to all atoms in the supercell. Source data are provided as a Source Data file.

## Precursor of $\beta$-Sn crystal

As seen from Supplementary Fig. 21, during the NG processes of $\beta$-Sn crystal from HDA, the $\beta$-Sn nuclei regions are always surrounded by atoms with large $\bar{q}_4^8$ or $\bar{q}_6^8$. This suggests that $\beta$-Sn crystals tend to be nucleated from high $\bar{q}_4^8$ or $q_6^8$ regions, which can be viewed as the precursors of $\beta$-Sn crystal. We note that a recent study[52] indicates that precursors also play an important role in crystal growth.

## Two-step formation of sh crystal

We found that the crystallisation of sh crystals from a-Si is also a two-step process, consistent with Ostwald's step rule[57]. Specifically, we revealed that sh crystal nuclei always emerge in $\beta$-Sn nuclei, and there is no sh-like atom before the emergence of $\beta$-Sn-like atoms in all crystallisation processes in the current work (see Figs. 4d, e and 5d as well as Supplementary Note 10). This non-classical transition pathway of the crystallisation process in a-Si has yet to be recognised in the recent work[33]. The reason why the $\beta$-Sn structure appears as an intermediate phase in the formation of the sh structure from a-Si should be that the orientational order of the $\beta$-Sn structure is intermediate between the amorphous forms and sh structure (see Supplementary Fig. 4a, b).

The observed process might appear to be a result of the 'continuous' evolution of local structures or the order parameter. However, this is not the case. Firstly, we emphasise that the distributions of the order parameters remain distinctly separated and maintain a significant distinction (refer to Supplementary Fig. 4a, b). Moreover, the identification of the crystallisation process leading to sh-crystals as a

manifestation of non-classical two-step nucleation is a conclusion drawn not only from the order parameter evolutions but also from spatial and temporal characteristics. As shown in Supplementary Fig. 15a, sh-crystals consistently emerge within pre-existing $\beta$-Sn crystal regions within HDA. Additionally, the surfaces of these crystals are consistently surrounded by $\beta$-Sn crystals. These observations offer clear indications of a two-step nucleation behaviour. This assertion gains further support from Supplementary Fig. 22, where the emergence of sh-crystals exclusively follows the formation of $\beta$-Sn crystals.

## Heating HDA and VHDA

We also heated the HDA$_{12,1000}$ and VHDA$_{15,10}$ samples at 12 and 15 GPa, respectively, from 300 K to 1500 K at a rate of $1 \times 10^{13}$ K s$^{-1}$. According to Fig. 6a, c, upon heating, the HDA-VHDA transition starts slightly earlier than the crystal nucleation. The fraction of VHDA-like atoms reaches its maximum of 47.1% at 573 K, at which the largest crystal nucleus contains 590 atoms. The crystal nuclei grow very quickly when further increasing temperature, and two crystalline grains, $\beta$-Sn and sh ones, form around 980 K. The sh crystal grain forms through a two-step process, i.e., $\beta$-Sn serves as an intermediate phase. A continuous temperature increase gradually transforms the sh crystal grain into the $\beta$-Sn structure before melting, indicating the higher stability of the $\beta$-Sn phase structure at higher temperatures. The partial transformation of HDA-like into VHDA-like atoms upon rapidly heating the HDA sample from 300 K to 500 K suggests that VHDA is more stable than HDA at 12 GPa and 500 K. This might explain why the LDA-HDA transition was not observed at 500 K[33].

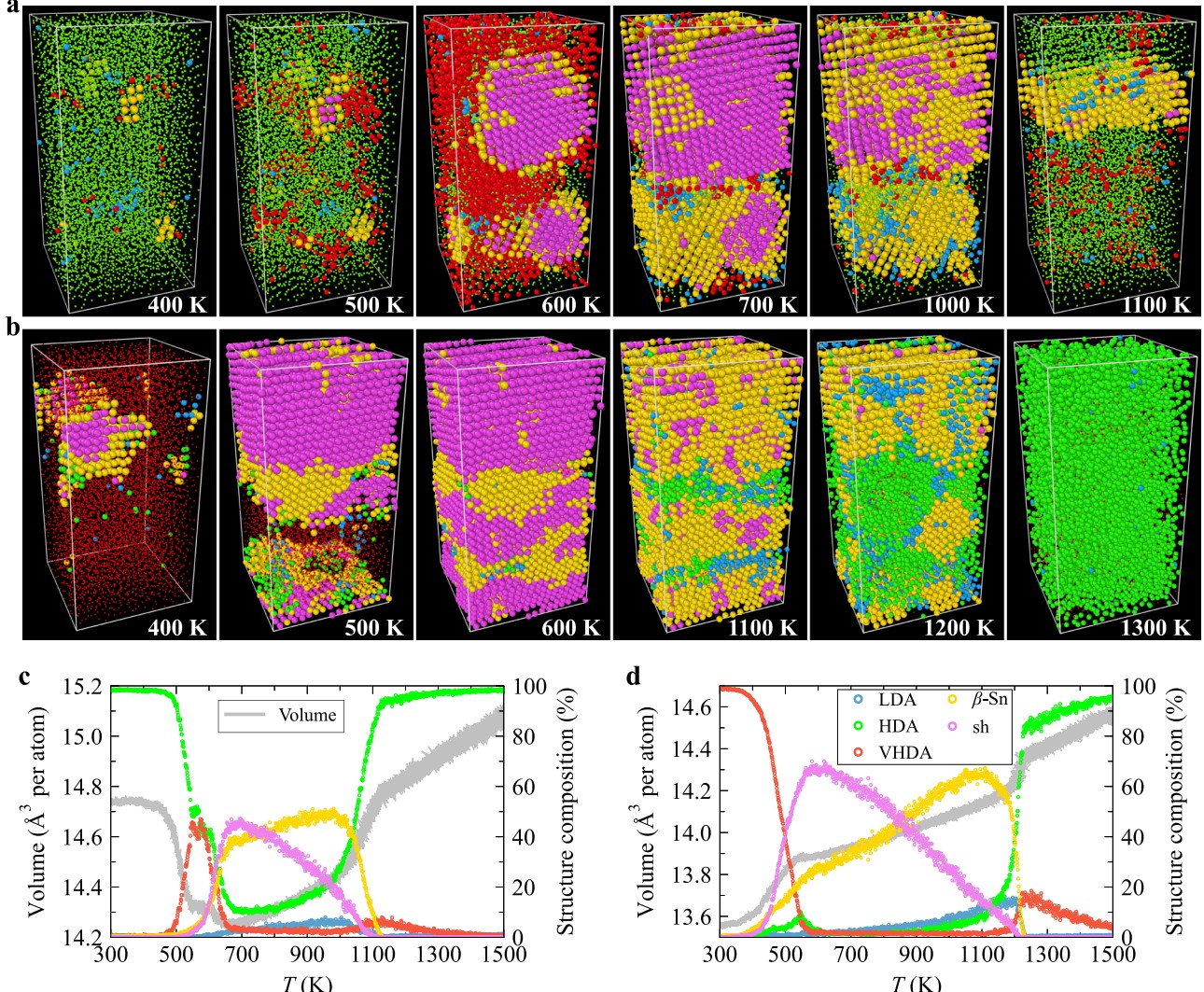

**Fig. 6 | The transformation of high-density amorphous (HDA) and very-high-density amorphous (VHDA) upon heating. a** The structural snapshots when heating the HDA$_{12,1000}$ sample at 12 GPa at a rate of $1 \times 10^{13}$ K s$^{-1}$. Blue, green, red, yellow, and magenta spheres represent low-density amorphous (LDA)-, HDA-, VHDA-, $\beta$-Sn-, and simple hexagonal (sh)-like atoms, respectively. The atom size is adjusted for different structural types for clarity. **b** The structural snapshots when heating the VHDA$_{15,10}$ sample at 15 GPa at a rate of $1 \times 10^{13}$ K s$^{-1}$. The colour scheme is the same as **a**. The evolution of the volume (left axis) and the fraction of different local environments (right axis) when heating the HDA$_{12,1000}$ sample at 12 GPa (**c**) and VHDA$_{15,10}$ sample at 15 GPa (**d**) from 300 to 1500 K at a rate of $1 \times 10^{13}$ K s$^{-1}$. Source data are provided as a Source Data file.

As seen from Fig. 6b, d, the VHDA first crystallises into a sh crystal through a two-step process, i.e., $\beta$-Sn serves as an intermediate phase when heating the VHDA$_{15,10}$ sample at 15 GPa, and the crystal also becomes more $\beta$-Sn-like at a higher temperature before finally melting into a liquid.

As can be seen from Fig. 6c, d as well as Supplementary Fig. 23, the liquid structures obtained at both 12 and 15 GPa are similar and closest to the HDA structure among the three amorphous forms but distinctly different from the liquid structure at 0 GPa. However, the presence or absence of LLT needs to be examined carefully.

## Outlook

In this study, we conducted an investigation into the structural transformations of amorphous silicon induced by rapid pressure changes. To accomplish this, we performed extensive molecular dynamics (MD) simulations over very long timescales, utilising a novel machine learning-based potential. Our research unveiled the presence of three distinct amorphous forms, namely, LDA, HDA, and VHDA, as well as two high-pressure crystalline forms in silicon, under varying pressure

conditions. These findings align closely with prior experimental observations[10,11] and ab-initio MD simulations[34,41].

We then disclosed the short-range orientational orders of the three amorphous forms and their relationships to the corresponding crystals, identifying the order parameters governing AAT and revealing a deep link between polyamorphism and polymorphism in the system. A summary figure is presented in Fig. 7. In addition, we have observed that, in comparison to both LDA and HDA, VHDA formed due to increased pressure exhibits a significantly greater degree of local structural disorder. This increased structural disorder renders VHDA less thermodynamically stable against crystallisation and also results in a lack of mechanical stability, as evidenced by the vibrational density of states.

We also followed the kinetics of AAT and crystallisation, with a particular focus on tracking the temporal evolution of local structures. In this context, we discerned two distinct types of structural transformations, categorised as NG and SD types. For the NG-type transformation, occurring as LDA transforms into HDA, we investigated the roughness of the interface within the nucleated domain. This

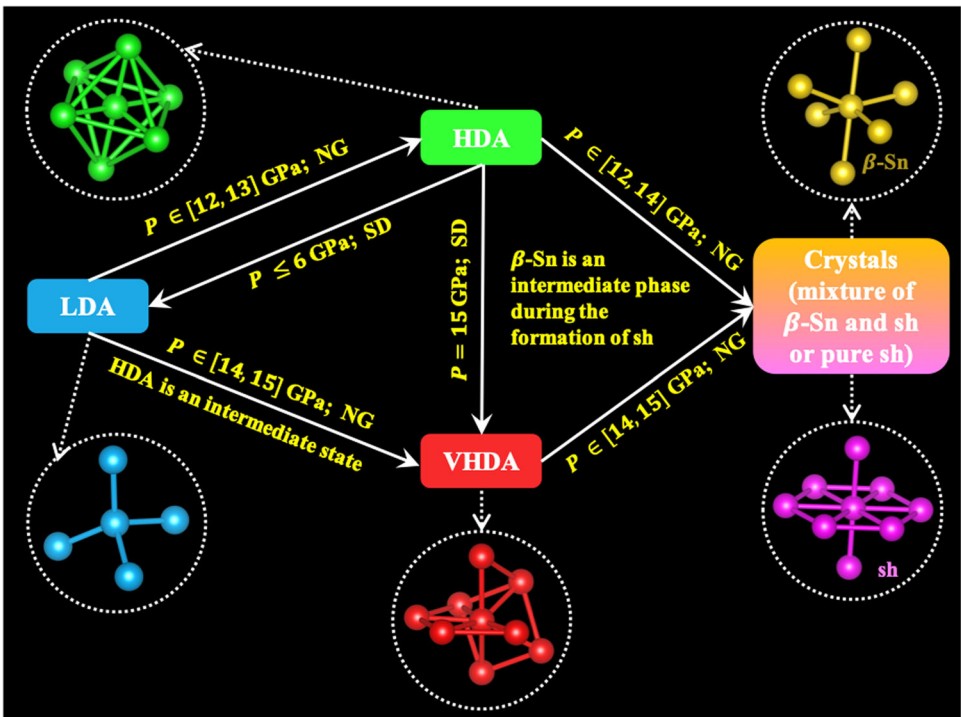

**Fig. 7 | Schematic Overview.** A schematic diagram outlining the structural transformation pathways among the three amorphous forms of a-Si, that is, low-density amorphous (LDA), high-density amorphous (HDA) and very-high-density amorphous (VHDA), along with their crystallisation routes to $\beta$-Sn and/or simple hexagonal (sh) structures, as elucidated in this study. The transition mechanisms, i.e, nucleation and growth (NG) and spinodal decomposition (SD), are marked for the transformations at different pressures ($P$). Additionally, the diagram presents representative local structures for the three amorphous forms and two crystal phases.

non-spherical shape of nuclei serves as a manifestation of the mechanical characteristics inherent in diffusion-less solid-state transformations. Furthermore, we found no direct transition path between LDA and VHDA; the transition must be through the intermediate HDA state. Similarly, the formation of sh crystal from amorphous states follows a two-step process involving an intermediate state known as the $\beta$-Sn crystal. These discoveries underscore the pivotal importance of preordering and mechanical instability in solid-state transformations, including both AAT and crystallisation processes. Preordering and mechanical instability play a crucial role in reducing the thermodynamic and mechanical barriers associated with these transformations.

Assessing the general applicability of these findings across various materials undergoing AATs and dissecting the distinct contributions of preordering and mechanical instability represent intriguing avenues for future research. Such investigations have the potential to shed further light on the fundamental principles governing structural transformations in diverse materials and provide valuable insights into how these factors can be manipulated or controlled in various applications.

## Methods
### Molecular dynamics simulations
We perform molecular dynamics (MD) simulations of the model silicon interacting with a recently developed ML potential[36], which is the same as the one used in ref. 33 and was also used to study the mechanical behaviour of a-Si recently[58]. The procedure of generating the as-quenched LDA form of a-Si is similar to that in refs. 33,58,59. Specifically, a supercell of cubic diamond silicon composed of $8(x) \times 8(y) \times 16(z)$ unit cells (thus, containing 8192 atoms) was heated to 2800 K at zero pressure and then equilibrated for 10 ps, followed by equilibration at 1800 K for 50 ps and then at 1500 K for 100 ps.

Subsequently, the liquid at 1500 K was first quenched to 1250 K at a rate of $1 \times 10^{13}$ K s$^{-1}$, followed by $1 \times 10^{11}$ K s$^{-1}$ to 1050 K, then at $1 \times 10^{13}$ K s$^{-1}$ to 300 K, and finally relaxed at 300 K for 100 ps. Then, the pressure was increased at various rates specified in the main text to various target values for performing isothermal-isobaric annealing to study different AAT and crystallisation processes. Unless otherwise stated, the temperature is always 300 K when applying hydrostatic pressure on the sample. We conducted all these quenching, pressurisation, and relaxation in the isothermal-isobaric ($NPT$) ensemble under a Nose–Hoover thermostat. The periodic boundary condition was applied in all three directions. The time step was always 1 fs. All MD simulations were implemented in the Large-scale Atomic/Molecular Massively Parallel Simulator (LAMMPS)[60] and the visualisation of atomic configurations was realised using the OVITO[61] and VESTA[62] packages. The initial and final configurations of molecular dynamics simulations for isothermal-isobaric relaxations at 10, 12, and 15 GPa are provided as a Supplementary Data file.

### Identifying local crystalline environments
We use coarse-grained local bond orientational order parameters $\bar{q}_l^N(i)$ to differentiate local crystalline environments from amorphous environments and separate $\beta$-Sn- and sh-like environments. First, a complex vector $q_{lm}(i)$ of atom $i$ is defined as[37]

$$q_{lm}(i) = \frac{1}{N} \sum_{j=1}^{N} Y_{lm}(\mathbf{r}_{ij}). \qquad (1)$$

Here, the constant $N$ determines how many nearest neighbours of atom $i$ will be taken into account, $l$ is a free integer parameter, and $m$ is an integer within the range of $[-l, l]$. $Y_{lm}(\mathbf{r}_{ij})$ is the spherical harmonic function, and $\mathbf{r}_{ij}$ is the vector connecting the central atom $i$ to its neighbouring atom $j$. Then, the coarse-grained local bond orientational

order parameters $\bar{q}_l^N(i)$ can be calculated as[38]

$$\bar{q}_l^N(i) = \sqrt{\frac{4\pi}{2l+1} \sum_{m=-l}^{l} |\bar{q}_{lm}(i)|^2}, \qquad (2)$$

where

$$\bar{q}_{lm}(i) = \frac{1}{N+1} \sum_{k=0}^{N} q_{lm}(k). \qquad (3)$$

Here, the summation runs over the first $N$ neighbours of atom $i$ and the atom $i$ itself.

To choose the best $\bar{q}_l^N$ which can accurately differentiate local crystalline environments from amorphous ones, we generated a set of snapshots for each of the three amorphous forms and two crystalline forms, i.e., $\beta$-Sn and sh crystals. Specifically,

(i) for LDA, we evenly extracted 21 snapshots when further relaxing the sample of $LDA_{10,200}$ denoted in Fig. 1a for 5 ps at 10 GPa and 300 K;

(ii) for HDA, we evenly extracted 21 snapshots when further relaxing the sample of $HDA_{12,1000}$ denoted on Fig. 1a for 5 ps at 12 GPa and 300 K;

(iii) for VHDA, we evenly extracted 21 snapshots when further relaxing the sample of $VHDA_{15,10}$ denoted in Fig. 1a for 5 ps at 15 GPa and 300 K;

(iv) for $\beta$-Sn, we created a bulk $\beta$-Sn crystal containing 4096 atoms, and after sufficient relaxation at 300 K and 12 GPa, we increased and decreased pressure on it from 12 to 15 and 0 GPa, respectively. 31 snapshots were chosen over the pressure range between 0 and 15 GPa (the pressure interval is 0.5 GPa);

(v) for sh, we created a bulk sh crystal containing 4096 atoms, and after sufficient relaxation at 300 K and 15 GPa, we reduced pressure on it from 15 to 10 GPa, during which 51 snapshots were extracted per 0.1 GPa. We found the sh crystal unstable when pressure is less than 10 GPa.

Note that we also considered other silicon crystal structures and found that the amount of local cubic diamond- or hexagonal diamond-like environments is negligible, and other crystal structures, such as Imma and Cmca, are not stable under the conditions we considered here.

Then, we calculated the distributions of $\bar{q}_l^N$ using many different combinations of $l$ and $N$ for the three amorphous forms, i.e., LDA, HDA, and VHDA, as well as the two crystals, i.e., $\beta$-Sn and sh, using these snapshots. As demonstrated in Supplementary Fig. 4a, the distributions of $\bar{q}_4^8$ for the three amorphous forms are well separated from those for the two crystals. We thus chose to use $\bar{q}_4^8 = 0.4$ as the threshold to differentiate crystal-like environments from amorphous-like environments. There is a small amount of overlap in the distribution of either $\bar{q}_4^8$ or $\bar{q}_6^8$ between $\beta$-Sn and sh crystals (see Supplementary Fig. 4a, b), we hence use the boundary in the $\bar{q}_4^8$-$\bar{q}_6^8$ plane (denoted as the red dashed line in Supplementary Fig. 4c) to separating sh-like environments from $\beta$-Sn-like ones.

**Distinguishing LDA-like environments from both HDA- and VHDA-like environments**
Separating different local amorphous-like environments is more challenging than differentiating different local crystal-like environments. To quickly choose the best $\bar{q}_l^N$ which can distinguish different local amorphous-like environments from a wide range of possible values for both $l$ and $N$, we use $O_{\alpha\beta}$ as defined in ref. 38 to quantify the degree of overlap in the distribution of a given $\bar{q}_l^N$ between two distinct structural forms, $\alpha$ and $\beta$. Here,

$$O_{\alpha\beta} = \frac{\int P_\alpha(x)P_\beta(x)dx}{\sqrt{\int P_\alpha^2(x)dx \int P_\beta^2(y)dy}}, \qquad (4)$$

where $P_\alpha(x)$ represents the probability density function of a given $\bar{q}_l^N$ for structural form $\alpha$. Obviously, a smaller value of $O_{\alpha\beta}$ means smaller overlap in the distribution of a given $\bar{q}_l^N$ between structural forms $\alpha$ and $\beta$.

As seen from Supplementary Fig. 4g, the lowest $O_{\text{LDA-HDA}}$ is 0.009 for $\bar{q}_4^{21}$ among all $\bar{q}_l^N$ with different values of $l$ and $N$. $\bar{q}_4^{21}$ is thus expected to separate LDA- and HDA-like environments well. Then, we plotted the distribution of $\bar{q}_4^{21}$ for all the three amorphous forms in Supplementary Fig. 4d. Visibly, the distribution of $\bar{q}_4^{21}$ for LDA has a tiny section overlapped with the corresponding distribution for HDA or VHDA form around the value of $\bar{q}_4^{21} = 0.03$. We hence determine amorphous-like environments with $\bar{q}_4^{21} > 0.03$ as LDA-like environments. With this threshold of $\bar{q}_4^{21}$, there are 98.1%, 0.8% and 0.3% of LDA-like environments in the $LDA_{10,200}$, $HDA_{12,1000}$ and $VHDA_{15,10}$ samples, respectively.

**Separating HDA- and VHDA-like environments**
We used the same strategy as above to search for the best $\bar{q}_l^N$, which can separate HDA- and VHDA-like environments. However, the lowest $O_{\text{HDA-VHDA}}$ is 0.363 for $\bar{q}_{10}^{13}$ among all $\bar{q}_l^N$ with different values of $l$ and $N$ (see Supplementary Fig. 4h). As shown in Supplementary Fig. 4e, there is indeed a large overlap in the distribution of $\bar{q}_{10}^{13}$ between HDA and VHDA samples. Table S1 lists the specific value of $N$, which can minimise $O_{\text{HDA-VHDA}}$ as well as the corresponding minimum value of $O_{\text{HDA-VHDA}}$ for each $l$ over the range from 1 to 15. We also tried the coarse-grained local bond orientational order parameter $\bar{w}_l^N$ (ref. 38) and found that the corresponding $O_{\text{HDA-VHDA}}$ of $\bar{w}_l^N$ is even much higher than that of $\bar{q}_l^N$. These results suggest that it is hard to well separate local HDA- and VHDA-like environments using a single coarse-grained local bond orientational order parameter $\bar{q}_l^N$ or $\bar{w}_l^N$. It may be because the resemblance between local HDA- and VHDA-like environments is non-trivial, and there is limited information contained in an individual $\bar{q}_l^N$ or $\bar{w}_l^N$ regarding the characteristics of local atomic environments. To better separate local HDA- and VHDA-like environments, we then trained a convolutional neural network (CNN) model[39] (a binary classifier) in which local atomic environments are described using spatial density maps (SDMs)[40,63]. The accuracy of the CNN model in separating VHDA-like environments from HDA-like environments can be as high as 99.66%, as shown in Supplementary Fig. 4f. The details regarding the construction of the CNN model can be found in the next section.

To show the advantage of the CNN model over other ML models in separating HDA- and VHDA-like environments, we also trained linear support vector machine (SVM) and conventional neural network (NN) models using multiple $\bar{q}_l^N$ as well as local radial density function $G_i(\bar{r})$[46,64] as structure representations. See below for the training procedures of these ML models. As shown in Supplementary Fig. 4f, the accuracy of the CNN model is much higher than all these SVM and NN models in separating HDA- and VHDA-like environments. It is interesting to note that the accuracy of the SVM model solely considering local angular information, i.e., multiple $\bar{q}_l^N$, is higher than that of the SVM model solely taking into account local radial density information, i.e., multiple $G_i(\bar{r})$ (see Supplementary Fig. 4f). This suggests that the difference in angular order is more pronounced than that in radial density order between HDA- and VHDA-like environments. The advantage of the CNN framework over other ML methods is due to (i) the completeness of the structure representation—SDM—used in the

CNN framework and (ii) the state-of-the-art learning capability of the CNN model[40,63,65].

### The details of the construction of the CNN model

**Atomic structure representation.** Similar to ref. 40, the atomic structure representation—SDM for a local atomic environment centred on atom $i$ is defined as

$$\Xi_i(\bar{x}, \bar{y}, \bar{z}) = \sum \exp\left(-\frac{(\bar{r}_{ij,x} - \bar{x})^2 + (\bar{r}_{ij,y} - \bar{y})^2 + (\bar{r}_{ij,z} - \bar{z})^2}{2\Delta^2}\right), \quad (5)$$

where the summation is performed over all neighbouring atoms with $|\mathbf{r}_{ij}| < \bar{r}_c$ as well as the central atom $i$ itself. To avoid the influence of pressure and/or temperature on bond lengths, here we use reduced vector $\bar{\mathbf{r}}_{ij} = \frac{\mathbf{r}_{ij}}{|\mathbf{r}_{i,m}|}$, where $|\mathbf{r}_{i,m}|$ is the shortest bond length of atom $i$. Here, $\bar{r}_{ij,x}$, $\bar{r}_{ij,y}$ and $\bar{r}_{ij,z}$ are the components along $x$, $y$ and $z$ directions of $\bar{\mathbf{r}}_{ij}$, respectively. $\bar{x}$, $\bar{y}$ and $\bar{z} \in [-l_c + 0.5\Delta, l_c - 0.5\Delta]$ with a constant increment of $\Delta$, where $\Delta$ determines the resolution of SDMs, and $l_c$ decides the size of SDMs and should be equal to or slightly larger than $\bar{r}_c$, which is the reduced cutoff determining the size of the local atomic environment. More discussion regarding choosing suitable values for the free parameters in SDM can be found in refs. 40,63. Here we chose $\bar{r}_c = 2.5$, $l_c = 3.0$ and $\Delta = 0.25$ after multiple trials. As shown in Supplementary Fig. 4i, there are around 40–75 and 50–80 neighbours in the local environments of HDA and VHDA, respectively, within a reduced cutoff $\bar{r}_c = 2.5$.

**CNN model architecture.** The SDMs generated with the parameters chosen above are three-dimensional (3D) numerical arrays, each containing $24 \times 24 \times 24$ elements, equivalent to 3D images, each containing $24^3$ voxels. Note that here each voxel only has one channel as there is only one species, i.e., silicon, in the materials we considered here, and more channels can be included in the SDM when there are more different species in the materials of interest. Therefore, we can directly feed these SDMs or images into a CNN model. For the binary classification task in the current case, i.e., judging whether a local atomic environment is HDA- or VHDA-like, we found a CNN model containing 4 convolutional layers is sufficient to achieve an accuracy very close to 100%. In our CNN model, 8 filters with a small receptive field of $3 \times 3 \times 3$ were used in each convolutional layer, and batch normalisation[66] was adopted right after each convolution and before activation with the rectification (ReLU) nonlinearity[67]. After activation, each of the first three convolutional layers was followed by a 3D max-pooling layer. Max-pooling is performed over a $2 \times 2 \times 2$ voxel window, with a stride of 2. The last convolutional layer is directly followed by the output layer, a single sigmoid neuron, as we perform a binary classification task.

**Training/test datasets.** To ensure that our CNN model is robust, we constructed a large test dataset containing 10,000 instances (5000 instances for each class). We found that a training dataset containing 10,000 instances for each class (20,000 instances in total) is large enough to train well this CNN model. Thus, we randomly chose ~477 and ~239 local environments from each of the 21 snapshots of the HDA$_{12,1000}$ sample to construct our training and test datasets, respectively. These local environments are labelled $y_i = 0$. Then, the equivalent numbers of local environments were selected randomly from the 21 snapshots of the VHDA$_{15,10}$ sample and added to the training and test datasets, respectively. These local environments are labelled $y_i = 1$. Since the SDM is rotationally non-invariant, we need to ensure that the prediction is always the same for a local environment at different orientations. Thus, we rotate each local environment such that its shortest bond to the central atom is parallel to the $x$ axis, and the first two nearest atoms and the central atom coexist in the $xy$ plane (the first three nearest atoms

will be required to coexist in the $xy$ plane if the first and second bonds are parallel to one another) before converting it into an SDM.

**Training procedure.** During the training of the CNN model, we minimised the binary cross-entropy loss between true and predicted labels, used early stopping, and selected the model with the highest accuracy on the test dataset. The learning rate started from 0.001 and was then divided by $\sqrt{10}$ once the test accuracy plateaued. We chose an RMSprop optimiser and used a mini-batch size of 300. We augmented the training data by randomly rotating each local environment before converting it into an image every time we fed it into the CNN. The training was implemented in the TensorFlow package[68].

### The training of the SVM and NN models

Firstly, we trained several linear support vector machine (SVM) models using a different number of the fifteen $\bar{q}_l^N$ listed in Supplementary Table 1 as structure representation. For example, including three $\bar{q}_l^N$ means that the first three $\bar{q}_l^N$ in Supplementary Table 1 were used. The variation of the accuracy of the SVM model with increasing the number of $\bar{q}_l^N$ used in the SVM model is exhibited in Supplementary Fig. 4f. Note that we also tried to add several $\bar{w}_l^N$ into the structure representation but found that including several $\bar{w}_l^N$ cannot improve the accuracy effectively.

We also used the Gaussian weighted local radial density function $G_i(\bar{r})$[46,64] as structure representation to train an SVM model. $G_i(\bar{r})$ is defined as

$$G_i(\bar{r}) = \sum_{\bar{r}_{ij} \le \bar{r}_c} \exp\left(-\frac{(\bar{r}_{ij} - \bar{r})^2}{2\Delta^2}\right), \quad (6)$$

where $\bar{r}_{ij} = |\bar{\mathbf{r}}_{ij}|$ and $\bar{r} \in [1.0 - 2\Delta, \bar{r}_c + 2\Delta]$ with a constant increment of $\Delta$. Here, we chose $\bar{r}_c = 2.5$, the same as in the CNN model, and found that a $\Delta = 0.1$ is sufficient. The accuracy of this SVM model is also described in Supplementary Fig. 4f.

In addition, we used multiple $\bar{q}_l^N$ in conjunction with $G_i(\bar{r})$ simultaneously as structure representation to train SVM models and used the fifteen $\bar{q}_l^N$ and $G_i(\bar{r})$ simultaneously as structure representation to train a conventional neural network (NN) model. Although the accuracy of these ML models can be increased when simultaneously considering both angular and radial density information or when a linear SVM model is replaced with a non-linear NN model, the highest accuracy achieved with the NN model is still lower than that of the CNN model (see Supplementary Fig. 4f).

The training of the SVM models was implemented in the Scikit-learn package[69]. The training procedure of the NN model is similar to those of the CNN models and implemented in the TensorFlow package[68]. We intensively optimised the architecture of the NN model and found that the optimal NN model contains two hidden layers, and each hidden layer contains 30 neurons activated with the ReLU nonlinearity[67], and a mini-batch size of 150 was used.

## Data availability
All study data are included in the article and Supplementary Information. Source data are provided with this paper.

## Code availability
The simulation codes used in this study are available from the corresponding authors upon request.

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

## Acknowledgements

Z.F. gratefully thanks En Ma for his thoughtful discussions and Gang Sun for his suggestions regarding local bond orientational order parameters and ring analysis. This work was partially supported by Specially Promoted Research (JP20H05619) from the Japan Society of the Promotion of Science (JSPS).

## Author contributions

Z.F. and H.T. conceived the project, and H.T. supervised the project. Z.F. performed research; Z.F. and H.T. analysed data and wrote the paper together.

## Competing interests

The authors declare no competing interests.
