## [Peer Review File · Nature Communications]

Microscopic mechanisms of pressure-induced amorphous-amorphous transitions and crystallisation in siliconREVIEWER COMMENTS

Reviewer #1 (Remarks to the Author):

In this paper the authors report their findings about the amorphous-amorphous phase transition using a ML based potential. The study have interesting and relevant findings about polyamorphous materials, and the manuscript can be accept with some minor revisions:

1 - The cutoff employed in the cluster analysis, 2.85 Å, is smaller than the second peak in the $g(r)$. Usually, materials with tetrahedral structure as silica, silicon and water have two characteristic length scales, then atoms in the same cluster may be arranged in such way that their distance in equal to the second length scale - or the second peak in the $g(r)$. In fact, a cluster analysis based purely in the inter particle bonding distance can be affected by the cutoff choice. The authors have tried another cutoff value, or just employed the same cutoff from the reference 32 without testing it?

2 - The authors evaluate the distribution of coarse-grained local bond orientational order parameter to characterize the phase transition. Recently, Hernandez and coauthors (J. Phys.: Condens. Matter 34 024002 2022) employed a similar method to analyze the amorphous-high density liquid transition in a core-softened model, and observed that using only the average quantity may lead to a not proper classification. The authors should add some discussion about this point.

Reviewer #2 (Remarks to the Author):

This manuscript provides a thorough exploration of the transition mechanisms between the different amorphous solid phases in a recently introduced machine-learning based model for silicon. In particular, the authors use extensive molecular dynamics simulations to observe the transition between the low-density amorphous (LDA), high-density amorphous (HDA), and very-high-density amorphous (VHDA) states at various pressures in the supercooled regime, and additionally observe crystallization into several different crystal structures. Their simulations elucidate the nature of the different phase transition processes (i.e. whether they occur via nucleation and growth or spinodal decomposition), and shed light on the competitions between mechanical stability and thermodynamic stability that govern phase transition in this non-equilibrium regime.

The study is carefully performed, with significant effort put into performing a careful analysis of the results using a suite of observables and structural order parameters to distinguish the different phases. The results are convincing and, in my opinion, provide valuable insights into the out-of-equilibrium phase behavior of an important system (even when considering that there are inevitably imperfections in any computational model adopted for simulation studies). In the cases where the transition processes are best described as nucleation and growth, there are clear waiting periods between the different transitions, giving strong weight to the interpretation of first-order transitions. Combined, the characterization of the model's phase behavior, the careful construction of methodology to distinguish different phases, and the insights into the transitions between different amorphous solid phases should appeal to a broad audience of researchers, and spark new studies. Additionally, the SI provides useful additional data for future investigations on this topic. Hence, I think the study is suitable for Nature Communications, and I am happy to recommend its publication. Below are some comments the authors may want to consider for a possible revision.

1) When considering nucleation and growth in a system without significant diffusion, one would expect that nucleation happens at a location in the system which is essentially chosen by the initial structure, rather than by random chance as when considering nucleation from a fluid. Do the authors know if, when running multiple simulations under the same conditions from the same starting point (e.g. when nucleating HDA from LDA), the nucleation process starts at the same location in the simulation box? Does this also mean that we would not expect nucleation times to be exponentially distributed when considering an ensemble of starting configurations (since, for example, some configurations might not have any sites that are likely to allow for nucleation on a reasonable simulation time scales, and others do)?

2) The authors label the emergence of the simple hexagonal structure as a form of non-classical nucleation, since the beta-Sn forms first. However, it also seems the beta-Sn is also located between the amorphous phases and the simple hexagonal phase (Supplemental Fig. 4). During the crystallization process, is there indeed a clear distinction between the two phases, or is it more of a continuum?

Reviewer #3 (Remarks to the Author):

This paper studies numerically the amorphous-amorphous transition of a computational model of silicon in non-equilibrium states.

In previous literature, the potential of the Stillinger-Weber model has been widely used. Although it demonstrated a liquid-liquid phase transition at equilibrium, it is known that the model does not describe high-pressure regions well. This article, therefore, uses a recently established model potential determined by a machine learning technique. The authors performed longer molecular simulations than previous studies with a protocol combining a ramp increase of pressure and subsequent relaxation at constant temperature and pressure. Using this non-equilibrium protocol, the authors characterized three structural states, LDA, HDA, and VHDA, on the basis of various structural indicators, including $g(r)$, bond orientation order parameters, and machine learning characterizations. The authors performed detailed simulations to reveal the kinetic pathways linking the different structural states.

This paper has carried out in-depth and extensive molecular simulations to clarify the microscopic features of the amorphous-amorphous transition in silicon, which should be appreciated. This is the main novelty. The authors have used the most advanced machine learning potential, although it has already been developed in previous literature. In addition, machine learning techniques for discerning different types of structures have already been established previously. In conclusion, the manuscript presents no technical novelties.

Overall, the manuscript is written in a plain manner or in the style of a regular article. The authors could therefore revise the manuscript in the style of a letter to attract a wider audience. For example, a series of concept panels summarizing the work could be added.

Nucleation and growth versus spinodal decomposition: Transition characterization based on these types is interesting. However, apart from the fact that it is only precisely defined in the mean-field case, the numerical methods for discerning these two types are not well described in this manuscript. For example, for the LDA-VHAD transition (page 9), it is not clear why one can conclude that it is an SD-type transition on the basis of Fig. 4b.

The authors compared the morphology in LLT and AAT. They wrote that while the LLT shown in Ref. [48] (real experiment) has a smooth profile, the AAT (this molecular simulation study) has a rough profile. But they belong to very different length scales. It is not certain that they can be compared.

Authors often mention mechanical characteristics in transition kinetics. However, there is no assessment of mechanical features in the main text or in the supplementary information. Thus argument is unclear.

Reviewer #4 (Remarks to the Author):

The present manuscript of Fan and Tanaka reports on the peculiar solid state amorphous-amorphous transitions and crystallization of amorphous silicon as studied by molecular dynamics simulations using a recently developed machine learning force-field. The authors study the impact of a number of simulation parameters (heating and pressurization rate) on the microscopic kinetics and morphology of the amorphous-amorphous and subsequent crystallization transitions. This is done through characterizing the atomic environments using classical bond order parameters as well as a neural network model to accurately separate and characterize them spatially.

While I think the work is thorough, sound, and of high quality, I am unsure if it is meeting the timely criteria of novelty for publication in Nature Communications, although it contributes nicely to the understanding of the rate effects and morphology of amorphous-amorphous transitions in silicon (and may perhaps guide the study of other amorphous materials).

For this reason, I recommend publication after transferring to a more dedicated journal, e.g., Communications Physics, as well as after addressing my various comments below.

Page 3, second paragraph: The authors state that the pressurization rate used by Deringer et al., (Nat. 589, 59, 2021) of 0.1 GPa ps⁻¹ is very high, and while it is indeed extreme compared to experiments, the authors themselves later (page 9) use a pressurization rate of up to 10 GPa ps⁻¹. Given that pressures often fluctuate in NPT simulations I would not consider a pressurization rate of 0.1 GPa ps⁻¹ to be extreme in simulations. I would suggest the authors to rephrase the sentence.

Page 3, second paragraph: The authors state "we conducted very long-timescale (relative to ab initio MD)....". While the timescale is indeed not within the timescale accessible reasonably for ab initio MD, the statement is somewhat vague. Please be more specific. Modern classical simulations may extend up to the microsecond scale and as these simulations are not comparable extreme, I feel the authors provide a wrong impression with the current phrasing.

Page 3, last paragraph: The authors write that "We jump pressure" and then in the next line "(a linear increase of P within 100 ps)". To me, a linear ramp of P is not jump pressurization. This "jump"-phrasing repeats other places in the manuscript.

Page 5-7: The authors introduce the concept of LDA, HDA, and VHDA phases. While there are significant differences between LDA and the other two, I feel that the authors could better support the existence of a discontinuous difference between the HDA and VHDA states. In fact, on pages 6-7 the authors state how "the main difference between HDA and VHDA is the fraction of atoms with different CN". Many amorphous phases see continuous CN changes as a function of pressure (for oxides see e.g., <https://doi.org/10.1063/5.0088606>). I urge the authors to better highlight why there is a structural difference between HDA and VHDA at the scale extending beyond the very short-range order. The authors may supplement the phrasing in the main manuscript with e.g., their ring analyses (among other) in Supplementary Figures 17-20.

Following the above point, and as also done in Ref. [32] of the manuscript, the authors may consider calculating the vibrational density of states for the various structural cases, especially

those with mixed phases. This should provide more in-depth knowledge on which dynamical differences exist between phases and may be a further support for the difference between HDA and VHDA phases.

Page 6, middle: "This suggests that the locally favoured structure of HDA has octahedral symmetry". How should one interpret this in terms of the clear presence of 60° angles in the bond angle distribution (BAD) in Fig. 1c? These would not be expected in the normal "perfect" octahedral case.

Page 8: "This provides more support to a positive answer...". While I think I understand the meaning of the sentence I encourage rephrasing and elaborating the sentence to make it more clear to the reader.

Page 8: "The 151 atoms which would form the HDA nucleus upon pressurization are indeed the atoms with relatively lower q_4^{-21} in the initial sample". This is a very interesting observation and I think the authors delve too little into this and related connections between initial structure and subsequent structural change upon pressurization. See my following points.

Although the authors do a somewhat extensive job describing the structural response under the various pressurization cases I suggest them to look deeper into the structural/dynamical origin of the points of nucleation. I imagine several ways to target this: 1) They could compute the eigenvectors using the dynamical matrix and from that gain fundamental knowledge of the vibrations of the atoms which later initiate phase change: Are the atoms vibrationally localized? Do nearby atoms move in or out of phase? 2) Given the authors interest in the use of machine learning the authors may look into the structural descriptor of so-called "softness" (see <https://doi.org/10.1038/nphys3644> and <https://doi.org/10.1103/PhysRevLett.122.028001>). Softness has so far seen very interesting usage for example to predict the nucleation sites in a simulated colloidal glass (<https://doi.org/10.1038/s41567-020-1016-4>). I wonder if softness could be able to predict the structural environments, the initiation sites for structural transitions, and perhaps even as a unified metric for the separation between LDA, HDA, and VHDA phases.

As the authors describe a lot of different pressurization and heating cases with varying structural output, it remains particularly hard to keep track of all these cases solely from the text. While the authors do briefly discuss some of my thoughts in Supplementary note 3 and Supplementary Figure 14, perhaps they could construct a type of "phase diagram" as a (supplementary?) figure to provide a general overview of which phases are present and stable under which conditions (P, rate of P, T).

Supplementary Figure 20a: Both the HDA and VHDA phases are seen to feature very abrupt changes in $g(r)$ (e.g., VHDA CN=6 and HDA CN=7). Do the authors believe this is physical or an artefact of the simulation/potential?

Supplementary Figure 20b: It is interesting to observe that in the VHDA phase, the lowest bond angles (~60°) are higher than in the HDA case. Perhaps the authors could dig into this effect by accessing the per atom stress in the two phases?

A point-by-point response to reviewers' comments

Responses to the comments of Reviewer #1

In this paper the authors report their findings about the amorphous-amorphous phase transition using a ML based potential. The study have interesting and relevant findings about polyamorphous materials, and the manuscript can be accept with some minor revisions:

Reply:

We thank the reviewer for carefully reading our manuscript and valuable comments. We are delighted to receive such a positive evaluation of our manuscript and are pleased to learn that the reviewer has recommended publication with some minor revisions.

1 - The cutoff employed in the cluster analysis, 2.85 Å, is smaller than the second peak in the $g(r)$. Usually, materials with tetrahedral structure as silica, silicon and water have two characteristic length scales, then atoms in the same cluster may be arranged in such way that their distance in equal to the second length scale - or the second peak in the $g(r)$. In fact, a cluster analysis based purely in the inter particle bonding distance can be affected by the cutoff choice. The authors have tried another cutoff value, or just employed the same cutoff from the reference 32 without testing it?

Reply:

In our manuscript, we utilized the identical cutoff as mentioned in reference 32. We thank the reviewer for highlighting the potential impact of selecting the cutoff on a cluster analysis solely based on inter-particle bonding distance. In order to ensure the robustness of our outcomes with respect to cutoff selection, we performed a thorough investigation, varying the cutoff value across a wide range.

As illustrated in Figure R1, the cluster analysis outcome remains consistent across a spectrum of cutoff values ranging from 2.85 to 3.72 Å, indicating its independence from the specific cutoff chosen. Notably, the second peak in the radial distribution function ($g(r)$) for the LDA and HDA samples occurs at approximately 3.72 Å and 3.54 Å, respectively. We have presented Fig. R1 as Supplementary Fig. 16, and explanatory notes (see the following sentence) regarding this have been incorporated into the captions of Figure 2 and on page 8 of the main text.

“In our cluster analysis, we employed a cutoff distance of 2.85 Å, consistent with the approach used in Ref. [33]. Importantly, we verified the robustness of our cluster analysis results by examining their independence from the chosen cutoff value within a range extending from 2.85 Å (corresponding to the first trough of $g(r)$) to 3.72 Å (corresponding to the second peak of $g(r)$ of LDA), as demonstrated in Supplementary Fig. 16.”

Fig. R1 The size evolution of the first largest HDA nuclei during the LDA-HDA transition when relaxing the as-quenched a-Si sample at 12 GPa. Cluster analysis was employed to ascertain the cluster size, wherein a range of cutoff values spanning from 2.85 to 3.72 Å (corresponding to the second peak in the $g(r)$ distribution of the LDA sample) were tested.

2 - The authors evaluate the distribution of coarse-grained local bond orientational order parameter to characterize the phase transition. Recently, Hernandez and coauthors (J. Phys.: Condens. Matter 34 024002 2022) employed a similar method to analyze the amorphous-high density liquid transition in a core-softened model, and observed that using only the average quantity may lead to a not proper classification. The authors should add some discussion about this point.

Reply:

We thank the reviewer for bringing the work by Hernandez et. al. to our attention. In response, we carefully reviewed their paper and formulated a potential explanation as to why they encountered challenges in distinguishing amorphous solids from high-density liquids using individual values of $q_l(i)$, $\bar{q}_l(i)$ or $\bar{\bar{q}}_l(i)$ in a core-softened model, although many different values for l in $q_l(i)$, $\bar{q}_l(i)$ and $\bar{\bar{q}}_l(i)$ were tested. It appears that their focus was predominantly on Voronoi tessellation as the sole method for determining nearest neighbours when computing $q_l(i)$, $\bar{q}_l(i)$ or $\bar{\bar{q}}_l(i)$, which could be a contributing factor to their results.

We have found in our current study that the number of neighbours N considered in the computation of $\bar{q}_l(i)$ also plays a significant role in determining the distribution of $\bar{q}_l(i)$ for different phases. This is exemplified by the distinct \bar{q}_4^8 and \bar{q}_4^{21} distributions exhibited by the various amorphous silicon forms, as highlighted in Supplementary Figure 4a and d. Through extensive exploration encompassing a wide range of values for both l and N , we have identified \bar{q}_4^{21} as the most effective metric for distinguishing LDA from the two higher-density amorphous silicon forms, as indicated in Supplementary Figure 4g. However, it is important to note that \bar{q}_l^N does not always serve as a reliable separator for various amorphous forms and liquid states. For example, we cannot find a single \bar{q}_l^N capable of

effectively distinguishing HDA from VHDA in silicon, as shown in Supplementary Figure 4e and h.

Following the suggestion of the reviewer, we have added the following sentences on page 9 of our revised manuscript.

“Here it is worth noting that when calculating the local bond orientational order parameter \bar{q}_l^N , the choice of the parameter l is not the only factor influencing its behaviour. The number of neighbouring atoms N also plays a pivotal role in determining the distribution of \bar{q}_l^N across various phases. For example, the distributions of \bar{q}_4^8 and \bar{q}_4^{21} for the three amorphous forms of silicon exhibit distinct differences, see Supplementary Fig. 4a, d. This aspect can potentially limit the effectiveness of \bar{q}_l^N in distinguishing between different phases, particularly when relying solely on a single method for determining neighbour lists.

A recent research [51] has pointed out that \bar{q}_l may not be consistently effective in distinguishing between an amorphous solid and a high-density liquid in a core-softened model. We hypothesise that the reason behind this limitation might be attributed to the sole use of Voronoi tessellation in constructing neighbour lists. Indeed, \bar{q}_l^N is not always effective in differentiating between various amorphous forms and/or liquid states. For instance, as detailed in Methods, identifying a single \bar{q}_l^N that reliably distinguishes HDA from VHDA in silicon proves to be a challenging task (Supplementary Fig. 4e, h).”

We hope that we have effectively addressed the reviewer's comments and that the revised manuscript is now well-suited for publication in Nature Communications.

Responses to the comments of Reviewer #2

This manuscript provides a thorough exploration of the transition mechanisms between the different amorphous solid phases in a recently introduced machine-learning based model for silicon. In particular, the authors use extensive molecular dynamics simulations to observe the transition between the low-density amorphous (LDA), high-density amorphous (HDA), and very-high-density amorphous (VHDA) states at various pressures in the supercooled regime, and additionally observe crystallization into several different crystal structures. Their simulations elucidate the nature of the different phase transition processes (i.e. whether they occur via nucleation and growth or spinodal decomposition), and shed light on the competitions between mechanical stability and thermodynamic stability that govern phase transition in this non-equilibrium regime.

The study is carefully performed, with significant effort put into performing a careful analysis of the results using a suite of observables and structural order parameters to distinguish the different phases. The results are convincing and, in my opinion, provide valuable insights into the out-of-equilibrium phase behavior of an important system (even when considering that there are inevitably imperfections in any computational model adopted for simulation studies). In the cases where the transition processes are best described as nucleation and growth, there are clear waiting periods between the different transitions, giving strong weight to the interpretation of first-order transitions. Combined, the characterization of the model's phase behavior, the careful construction of methodology to distinguish different phases, and the insights into the transitions between different amorphous solid phases should appeal to a broad audience of researchers, and spark new studies. Additionally, the SI provides useful additional data for future investigations on this topic. Hence, I think the study is suitable for Nature Communications, and I am happy to recommend its publication. Below are some comments the authors may want to consider for a possible revision.

Reply:

We thank the reviewer for carefully reading our manuscript and valuable comments. We are delighted to receive such a positive evaluation of our manuscript and are pleased to learn that the reviewer has recommended publication with some revisions.

1) When considering nucleation and growth in a system without significant diffusion, one would expect that nucleation happens at a location in the system which is essentially chosen by the initial structure, rather than by random chance as when considering nucleation from a fluid. Do the authors know if, when running multiple simulations under the same conditions from the same starting point (e.g. when nucleating HDA from LDA), the nucleation process starts at the same location in the simulation box? Does this also mean that we would not expect nucleation times to be exponentially distributed when considering an ensemble of starting configurations (since, for example, some configurations might not have any sites that are likely to allow for nucleation on a reasonable simulation time scales, and others do)?

Reply:

We fully agree that this is a very interesting point. As stated in the last paragraph in Supplementary Note 7 of our initial manuscript, “Interestingly, the HDA nucleus (95 atoms) at 12.5 GPa (case 1) after relaxation for 20 ps and the nucleus (151 atoms) at 12 GPa after relaxation for 100 ps commonly share 61 atoms in the initial as-quenched sample, suggesting the HDA nucleation initiated approximately in the same location at 12 and 12.5 GPa. This is further evidence of the significant impact of preordering on the HDA nucleation.” This observation supports the scenario that “*nucleation happens at a location in the system which is essentially chosen by the initial structure, rather than by random chance as when considering nucleation from a fluid*”. Now we have moved these sentences into the main text of our revised manuscript to highlight this interesting result.

As part of our revision process, we undertook two additional simulations to bolster this assertion. In the two simulations, the pressure (P) applied to the same initial a-Si sample at 300 K was instantaneously increased from 0 to 12 GPa, following the assignment of different initial velocity fields. This was followed by an isothermal-isobaric relaxation lasting 300 ps. The two simulations are different from our previous simulation where we applied a linear pressure ramp from 0 to 12 GPa. Notably, HDA nuclei of 151 atoms in these three simulations share 46 common atoms, as can be seen in Fig. R2. This outcome serves to validate the proposition that the nucleation sites governing the AAT in silicon at room temperature are fundamentally dictated by the initial structural configuration, i.e., preexisting preorder.

It is also interesting to explore whether the incubation period before nucleation in the AAT follows an exponential distribution. However, addressing this question necessitates the quenching of numerous independent samples, a task demanding extensive simulation durations. More specifically, the utilization of the present machine learning potential led to a substantial time requirement of nearly three months for effectively quenching an a-Si model. Consequently, the execution of extended-duration isothermal-isobaric relaxation simulations aimed at inducing HDA nucleation within the LDA context demands even greater time commitments. We thus leave it for future investigation.

We have added these new results and the related discussions (see the description from the bottom of page 10 to the top of page 11). And Fig. R2 was added as Supplementary Fig. 18.

“Regarding the influence of preordering on AAT, it is worth noting that the LDA-HDA transitions observed at both 12.5 and 13 GPa for the same initial state exhibit similarities to those previously discussed (please refer to Supplementary Fig. 17 and additional discussion in Supplementary Note 7). Interestingly, both the HDA nucleus containing 95 atoms at 12.5 GPa (case 1) after relaxation for 20 ps and the nucleus with 151 atoms at 12 GPa after relaxation for 100ps share a common set of 61 atoms that were part of the initial as-quenched sample. This commonality suggests that the initiation of HDA nucleation occurred in approximately the same location at both 12 and 12.5 GPa,

providing further evidence for the substantial influence of preordering on HDA nucleation. In other words, this observation implies that nucleation sites in solid-state phase transformations are determined by the initial solid structure, as opposed to the stochastic nature of nucleation from a fluid.

To further validate this notion, we conducted two additional simulations, each involving an instantaneous jump in P from 0 to 12 GPa on the same initial a-Si sample at 300 K. These simulations were preceded by the assignment of distinct initial velocity fields. Subsequently, an isothermal-isobaric relaxation spanning 300 ps was performed. Remarkably, the first HDA nuclei consisting of 151 atoms, observed across both cases, and even during relaxation following a linear pressure ramp from 0 to 12 GPa, commonly share 42 atoms, as depicted in Supplementary Fig. 18. This result strongly supports the conclusion that nucleation sites during AAT in silicon at room temperature are indeed determined by the initial structure. Furthermore, investigating whether the incubation period preceding nucleation in AAT adheres to an exponential distribution presents an intriguing avenue. However, addressing this question would require quenching multiple independent samples, which is computationally demanding. Therefore, we leave it for future studies.”

Fig. R2 The snapshot of the initial a-Si sample at 0 GPa and 300 K. The atoms that later form the nuclei of HDA structures are depicted across three distinct simulations. These simulations use the same initial state but employ different pressure increase protocols, including a linear pressure ramp from 0 to 12 GPa over 100 ps, as well as two instantaneous pressure jumps from 0 to 12 GPa, each initiated with a different set of initial velocity distributions. Within the context of the three simulations, the pink spheres represent the 42 atoms consistently present within the 151-atom HDA nuclei. These nuclei are shared among all three simulations, despite their different pressure increase protocols. Cyan, lime, and yellow spheres denote the remaining 109 atoms forming the 151-atom HDA nuclei in the three simulations, respectively. Small white spheres indicate

atoms that do not participate in the composition of the 151-atom nuclei in any of the three cases. The atom sizes have been adjusted to enhance clarity.

2) The authors label the emergence of the simple hexagonal structure as a form of non-classical nucleation, since the β -Sn forms first. However, it also seems the β -Sn is also located between the amorphous phases and the simple hexagonal phase (Supplemental Fig. 4). During the crystallization process, is there indeed a clear distinction between the two phases, or is it more of a continuum?

Reply:

As noted by the reviewer, the bond-orientational order parameters of β -Sn occupy a position intermediate to that of the amorphous phases and sh-crystals, as depicted in Supplementary Fig. 4. Nevertheless, their distributions maintain a noticeable distinction and are well-separated. We identify the crystallisation process leading to sh-crystals as a form of non-classical two-step nucleation. This conclusion arises not solely from the order parameter evolutions but also from the spatial and temporal characteristics. As shown in previous Supplementary Fig. 9a or current Supplementary Fig. 15a, sh-crystals consistently emerge within already-formed β -Sn crystal regions existing within HDA. Moreover, the surface of the crystal is consistently enveloped by β -Sn crystals. These observations provide clear indications of a two-step nucleation behaviour. This assertion gains further support from Fig. R3a and b-e, where the appearance of sh-crystals is exclusively subsequent to the formation of β -Sn crystals.

Fig. R3 A two-step crystallisation process. **a** shows the temporal change of the fraction of β -Sn- and sh-like local environments when annealing LDA sample at 15 GPa (case 1). **b-e** show the order-parameter distributions of local crystalline environments on the \bar{q}_4^8 - \bar{q}_6^8 plane at four time-windows (denoted on each panel) for the crystallisation process shown in **a**. Red (blue) colour corresponds to high (low) density.

We have added the following explanation on page 14.

“The observed process might appear to be a result of the 'continuous' evolution of local structures or the order parameter. However, this is not the case. Firstly, we emphasize that the distributions of the order parameters remain distinctly separated and maintain a significant distinction (refer to Supplementary Fig. 4a, b). Moreover, the identification of the crystallisation process leading to sh-crystals as a manifestation of non-classical two-step nucleation is a conclusion drawn not only from the order parameter evolutions but also from spatial and temporal characteristics. As shown in Supplementary Fig. 15a, sh-crystals consistently emerge within pre-existing β -Sn crystal regions within HDA. Additionally, the surfaces of these crystals are consistently surrounded by β -Sn crystals. These observations offer clear indications of a two-step nucleation behaviour. This assertion gains further support from Supplementary Fig. 22, where the emergence of sh-crystals exclusively follows the formation of β -Sn crystals.”

We hope that we have effectively addressed the reviewer's comments and that the revised manuscript is now well-suited for publication in Nature Communications.

Responses to the comments of Reviewer #3

This paper studies numerically the amorphous-amorphous transition of a computational model of silicon in non-equilibrium states. In previous literature, the potential of the Stillinger-Weber model has been widely used. Although it demonstrated a liquid-liquid phase transition at equilibrium, it is known that the model does not describe high-pressure regions well. This article, therefore, uses a recently established model potential determined by a machine learning technique. The authors performed longer molecular simulations than previous studies with a protocol combining a ramp increase of pressure and subsequent relaxation at constant temperature and pressure. Using this non-equilibrium protocol, the authors characterized three structural states, LDA, HDA, and VHDA, on the basis of various structural indicators, including $g(r)$, bond orientation order parameters, and machine learning characterizations. The authors performed detailed simulations to reveal the kinetic pathways linking the different structural states.

This paper has carried out in-depth and extensive molecular simulations to clarify the microscopic features of the amorphous-amorphous transition in silicon, which should be appreciated. This is the main novelty.

Reply:

We thank the reviewer for carefully reading our manuscript and valuable comments. We are delighted to receive a positive evaluation of our manuscript.

The authors have used the most advanced machine learning potential, although it has already been developed in previous literature. In addition, machine learning techniques for discerning different types of structures have already been established previously. In conclusion, the manuscript presents no technical novelties.

Reply:

While we admit that the potential we employed, as well as the use of machine learning for discerning distinct local structures, might not be considered novel, we firmly assert that our study introduces novelty in several aspects. Notably, our accomplishment in accurately identifying all relevant locally favoured structures within three distinct amorphous states, in addition to relevant high-pressure crystalline forms, constitutes a novel achievement. Furthermore, the elucidation of the intricate non-equilibrium kinetic pathway spanning multiple structural order parameter spaces during amorphous-amorphous transitions serves as a noteworthy contribution. We hope the reviewer will appreciate the significance of these points.

Overall, the manuscript is written in a plain manner or in the style of a regular article. The authors could therefore revise the manuscript in the style of a letter to attract a wider audience. For example, a series of concept panels summarizing the work could be added.

Reply:

While our paper is not specifically categorized as a Letter, we acknowledge the value of presenting a concise overview of our primary findings. Also incorporating the suggestion of Reviewer #4, we have included a summary diagram as Fig. 7 in our revised manuscript, which is also shown below as Fig. R4.

Fig. R4 A schematic diagram illustrating the structural transformation paths among the three amorphous forms of a-Si and their crystallisation paths investigated in the current work. The typical local structures of the three amorphous forms and two crystals of silicon are also shown.

Nucleation and growth versus spinodal decomposition: Transition characterization based on these types is interesting. However, apart from the fact that it is only precisely defined in the mean-field case, the numerical methods for discerning these two types are not well described in this manuscript. For example, for the LDA-VHAD transition (page 9), it is not clear why one can conclude that it is an SD-type transition on the basis of Fig. 4b.

Reply:

We apologize for any confusion caused by our previous statement. We used the term "SD-like manner" to describe the immediate appearance of the intermediate state—HDA without an evident incubation period in the LDA-VHAD transition. However, as the Reviewer pointed out, we acknowledge that this wording was misleading. To address this concern, we have rephrased it as "very quickly (~2 ps) without an evident incubation period", see page 13. Actually, we believe the LDA-VHAD transition occurs via an NG

process, according to Fig. 4a. The rapid onset of the transformation can be attributed to the presence of a precursor that triggers the process. In the section of “HDA-VHDA transition”, there was a sentence “..., unlike the NG process of forming the VHDA form in the LDA-VHDA transition at the same pressure (see above)”.

To make this point clearer, we have added the following sentence at the end of the “LDA-VHDA transition” section of the revised manuscript:

“As can be seen from Fig. 4a, the formation of VHDA in the LDA-VHDA transition appears to be an NG process. The rapid onset of the LDA-VHDA transition is catalyzed by the intermediate HDA form.”

We appreciate the reviewer for bringing this issue to our attention.

The authors compared the morphology in LLT and AAT. They wrote that while the LLT shown in Ref. [48] (real experiment) has a smooth profile, the AAT (this molecular simulation study) has a rough profile. But they belong to very different length scales. It is not certain that they can be compared.

Reply:

We completely agree with the reviewer's assessment. In our revised manuscript, we have eliminated the macroscopic observation of the liquid-liquid transition as a means to demonstrate the smoothness of the droplet interface in LLT. Instead, we have conducted a quantitative analysis of the nucleus shape to provide a more accurate representation of our findings, which is presented in Fig. R5.

Fig. R5 The relationship between the surface area-to-volume ratio of the largest HDA nucleus and its radius (derived from the volume, assuming nuclei are spherical) during the initial 400 ps of relaxation of the as-quenched a-Si sample at 12 GPa, subsequent to a linear pressure ramp within 100 ps. The black dashed curve represents the surface area-to-volume ratio of an ideal sphere ($3/r$) as a function of r . The surface area of the HDA nuclei was computed using the Gaussian blurring of the density field, while the volume was determined by summing the atomic volumes of the constituent atoms within

the HDA nuclei. This atomic volume calculation was made through Voronoi analysis. These calculations were performed using the OVITO software.

We also examined the displacement fields in the nucleation-growth process of HDA (see Fig. R6). We cannot see any coherent displacement to reduce the interfacial area of HDA domains, reflecting the non-diffusive nature of solid-state transformations.

Fig. R6 The projection on the XZ (a) and YZ (b) planes of the displacement field during the HDA nucleation-growth process (the atomic positions of the snapshot at 100 ps minus those of the snapshots at 0 ps during the isothermal-isobaric relaxation at 12 GPa). Both the width and transparency of the displacement vectors are scaled with its length. The largest displacement over the 100 ps is 3.94 Å.

We have added the following sentences in the revised manuscript (page 11):
“The non-spherical nature of an HDA nucleus becomes evident through the analysis of the surface area-to-volume ratio as it evolves (Supplementary Fig. 19a). Notably, there is no consistent trend of the nucleus developing a spherical shape over time. This observation emphasizes the combined influence of both thermodynamics and mechanics during the growth process. It is important to note that the interface energy does not play

a central role in this process, as the transformation is driven by non-diffusive mechanical collapse rather than thermal diffusion. This observation is further supported by the displacement fields in the nucleation-growth process of HDA (see Supplementary Fig. 19b). These displacement fields lack coherent motion aimed at minimizing the interfacial area of HDA domains. These findings collectively highlight the distinctive characteristics of non-diffusive solid-state transformations [55,56].”

Authors often mention mechanical characteristics in transition kinetics. However, there is no assessment of mechanical features in the main text or in the supplementary information. Thus argument is unclear.

Reply:

We acknowledge that our previous explanation of the term "mechanical features" was unclear. In order to provide a more accurate description, we have described that this term encompasses phenomena unique to solid-state transformations, such as diffusionless transformations and the mechanical collapsing of structural motifs with large volumes.

Specifically, we have added the following sentences at the last of the Introduction:
“Furthermore, our findings offer insights into the mechanistic contributions that arise from diffusionless solid-state transformations, a phenomenon that can occur due to processes such as the mechanical collapse of structural motifs characterized by significant local volume reduction.”

Please also see our reply to the preceding comment.

Additionally, we have conducted calculations to determine the atomic-level pressure changes during an LDA-to-HDA transition, as illustrated in the figure below (Fig. R7). Notably, the slight pressure difference between these two phases and the broad atomic-level pressure distribution even after coarse-graining constitute distinctive hallmarks of solid-state transformations, distinguishing them from their liquid-state counterparts. We can also see a heterogeneous spatial distribution of the coarse-grained pressure in Fig. R8. We have included these figures in Supplementary Fig. 19.

We have also added the following sentences to the main text of the revised manuscript (page 12):

“In addition, we evaluated the atomic-level pressure changes during an LDA-HDA transition, as illustrated in Supplementary Fig. 19c. It is noteworthy that a subtle pressure contrast between the two phases is observable, along with a widespread atomic-level pressure dispersion even after undergoing a coarse-graining process. Moreover, we have observed a non-uniform spatial distribution of the pressure in its coarse-grained form, as highlighted in Supplementary Fig. 19d. These results together serve as distinctive indicators of the mechanical characteristics inherent in solid-state transformations, a trait that is absent in their counterparts in the liquid state.”

Fig. R7 The distribution of atomic-level pressure after conducting coarse-grained up to neighbour atoms three times, at four time points (denoted on each panel) during the isothermal-isobaric relaxation at 12 GPa after a linear pressure ramp within 100 ps.

Fig. R8 The spatial distribution of the atomic-level pressure (coarse-grained up to neighbour atoms three times to reduce the noise) at 400 ps corresponding to the one in Fig. 2b. The small and large spheres correspond to LDA- and HDA-like atoms, respectively.

We hope that we have effectively addressed the reviewer's comments and that the revised manuscript is now well-suited for publication in Nature Communications.

Responses to the comments of Reviewer #4

The present manuscript of Fan and Tanaka reports on the peculiar solid state amorphous-amorphous transitions and crystallization of amorphous silicon as studied by molecular dynamics simulations using a recently developed machine learning force-field. The authors study the impact of a number of simulation parameters (heating and pressurization rate) on the microscopic kinetics and morphology of the amorphous-amorphous and subsequent crystallization transitions. This is done through characterizing the atomic environments using classical bond order parameters as well as a neural network model to accurately separate and characterize them spatially.

While I think the work is thorough, sound, and of high quality, I am unsure if it is meeting the timely criteria of novelty for publication in Nature Communications, although it contributes nicely to the understanding of the rate effects and morphology of amorphous-amorphous transitions in silicon (and may perhaps guide the study of other amorphous materials).

For this reason, I recommend publication after transferring to a more dedicated journal, e.g., Communications Physics, as well as after addressing my various comments below.

Reply:

We extend our gratitude to the reviewer for their thorough evaluation of our manuscript and the valuable comments provided. We are delighted to receive a positive assessment of our work. However, we respectfully disagree with the reviewer's opinion that our manuscript does not meet the criteria for publication in Nature Communications.

Our work offers significant novelty in several aspects. Firstly, we have utilized a recently-developed state-of-the-art machine learning potential to accurately describe the high-pressure phase behaviour of silicon. This represents a cutting-edge approach in the field. Additionally, we have conducted extensive simulations using this model, combined with a novel rapid pressure increase method to investigate the non-equilibrium phase transformation process. This rapid-pressure change protocol, which is generally assumed in the theoretical description of phase-transition dynamics, sheds light on a previously unexplored nonequilibrium aspect of the system at high pressures and enhances our understanding of the transformation kinetics of AAT.

In terms of structural analysis, we have successfully identified all pertinent locally favoured structures in three distinct amorphous states and their corresponding high-pressure crystals. To achieve this, we have combined bond-orientational order parameters, other structural measures, and machine learning techniques. This comprehensive approach provides a robust framework for studying structural properties.

Moreover, our significant contributions lie in elucidating the intricate non-equilibrium kinetic pathway across multiple structural order parameter spaces during amorphous-amorphous transitions, as well as discussing the mechanical characteristics of these transitions. This deeper understanding of the system's transformation processes adds to the existing body of knowledge in the field.

Overall, these findings represent novel and valuable contributions to the study of high-pressure phase behaviour and the solid-state phase transformation kinetics in silicon. We sincerely hope that the reviewer recognizes the importance and significance of these achievements.

Page 3, second paragraph: The authors state that the pressurization rate used by Deringer et al., (Nat. 589, 59, 2021) of 0.1 GPa ps^{-1} is very high, and while it is indeed extreme compared to experiments, the authors themselves later (page 9) use a pressurization rate of up to 10 GPa ps^{-1} . Given that pressures often fluctuate in NPT simulations I would not consider a pressurization rate of 0.1 GPa ps^{-1} to be extreme in simulations. I would suggest the authors to rephrase the sentence.

Reply:

Following the suggestion, we have rephrased the sentence to *“the continuous pressurisation, sustained at a rate of 0.1 GPa/ps ”*.

Page 3, second paragraph: The authors state “we conducted very long-timescale (relative to ab initio MD)...”. While the timescale is indeed not within the timescale accessible reasonably for ab initio MD, the statement is somewhat vague. Please be more specific. Modern classical simulations may extend up to the microsecond scale and as these simulations are not comparable extreme, I feel the authors provide a wrong impression with the current phrasing.

Reply:

Following the suggestion, we have rephrased the sentence as follows:

“One potential explanation for why Deringer et al. did not observe a complete transition between LDA and HDA [33] could be attributed to the continuous pressurization, sustained at a rate of 0.1 GPa/ps , which was utilized in their simulations. In contrast, our approach involves a different pressure elevation protocol that is better suited for investigating the kinetics of phase transformations. To be more precise, we executed a rapid linear pressure increase on the a-Si sample, raising it to a desired level within the range of 10 to 15 GPa at 300 K within 100 ps. This can be approximated as an almost instantaneous alteration in pressure. Subsequently, we followed the relaxation process for up to 3 ns under isothermal-isobaric conditions.”

We note that the longest isothermal-isobaric relaxation we conducted is indeed 3 ns, please see Supplementary Fig. 3a, although the time scale shown in Fig. 1a is 2 ns. Despite the appearance of a relatively short simulation period, it is important to note that the duration is actually quite substantial, particularly when considering the computational cost associated with simulations employing machine-learning potentials.

Page 3, last paragraph: The authors write that “We jump pressure” and then in the next line “(a linear increase of P within 100 ps)”. To me, a linear ramp of P is not jump pressurization. This “jump”-phrasing repeats other places in the manuscript.

Reply:

In the revision, we replaced “jump pressure” with “rapid increase pressure” throughout the manuscript when pressure was ramped linearly or at a constant rate. It is worth noting, however, that certain instances where pressure was instantaneously changed were retained as “jump” in the manuscript.

Page 5-7: The authors introduce the concept of LDA, HDA, and VHDA phases. While there are significant differences between LDA and the other two, I feel that the authors could better support the existence of a discontinuous difference between the HDA and VHDA states. In fact, on pages 6-7 the authors state how “the main difference between HDA and VHDA is the fraction of atoms with different CN”. Many amorphous phases see continuous CN changes as a function of pressure (for oxides see e.g., <https://doi.org/10.1063/5.0088606>). I urge the authors to better highlight why there is a structural difference between HDA and VHDA at the scale extending beyond the very short-range order. The authors may supplement the phrasing in the main manuscript with e.g., their ring analyses (among other) in Supplementary Figures 17-20.

Reply:

We agree that there is a structural difference between HDA and VHDA at the scale extending beyond the very short-range order.

Following the suggestion, we have changed this part as below (page 7):

“We also found that the local atomic environments centred on atoms with the same CN are comparable between HDA and VHDA (see Supplementary Note 6). This suggests that the primary difference within the first neighbouring shell between HDA and VHDA lies in the fraction of atoms with different CN. This observation supports a multiple-order-parameter model of LLT and AAT [8,22,41]. It is worth noting that, according to this model, two structural order parameters, in addition to the density order parameter describing the gas-liquid transition, are necessary to account for the existence of three distinct amorphous states [8,22]. Additionally, apart from the pronounced difference in CN, there are structural distinctions within the intermediate range between HDA and VHDA forms. This is supported by the radial distribution function, $g(r)$ (depicted in Fig. 1b), as well as

the ring analysis (Supplementary Fig. 10). Furthermore, there are interesting distinctions among the three amorphous states in terms of the density difference between an amorphous state and its corresponding crystalline counterpart and the shift in the first peak of $g(r)$ with increasing pressure, see Supplementary Fig. 11 and Supplementary Note 4.”

Following the above point, and as also done in Ref. [32] of the manuscript, the authors may consider calculating the vibrational density of states for the various structural cases, especially those with mixed phases. This should provide more in-depth knowledge on which dynamical differences exist between phases and may be a further support for the difference between HDA and VHDA phases.

Reply:

Following the reviewer’s advice, we have computed the vibrational density of states (VDOS) for the three amorphous states by conducting Fourier transformations of the velocity auto-correlation functions within these states. Evidently, there exists a noticeable disparity in VDOS between LDA and HDA. Conversely, the differentiation between HDA and VHDA is rather subtle (Fig. R9a). Nonetheless, upon normalizing the VDOSs by the Debye level, a noticeable discrepancy becomes evident in the low-frequency range: VHDA lacks the characteristic boson peak present in HDA and instead displays a strong quasi-elastic component (Fig. R9b). This observation suggests the intrinsically unstable nature of VHDA, aligning with its susceptibility to crystallisation.

Fig. R9 The vibrational density of states (VDOS) (a) and reduced VDOS (b) for the three different amorphous forms. These curves were obtained through Fourier transformations of the velocity auto-correlation functions (VACFs). To ensure higher accuracy, we averaged 21 VACF curves over a given time window for each of the three glass forms. Specifically, for LDA, we evenly chose 21 states over the time interval of 200-400 ps during relaxation at 10 GPa after a linear pressure increase; for HDA, we used 21 states over the time interval of 1000-1200 ps during relaxation at 12 GPa after a linear pressure

increase. The corresponding volume evolution for the two time windows are shown in Fig. 1a. For VHDA, we used 21 states over the time interval of 30-50 ps during relaxing LDA_{10,200} at 15 GPa (case 1) after a quick pressure increase. The corresponding volume evolution for this time window is shown in Supplementary Fig. 1a.

Fig. R10 The temporal evolution of the vibrational density of states (VDOS) (a) and reduced VDOS (b) for HDA during relaxation at 12 GPa after a linear pressure increase. For each time window shown in the legend, 10 velocity auto-correlation functions (VACFs) were averaged before conducting Fourier transformations. In b, we can see that the quasi-elastic component decreases with time, indicating the stabilization of HDA during the relaxation process.

We have added the following sentences in the revised manuscript (page 7):

“Moreover, we calculated the vibrational density of states (VDOS) for each amorphous form using the Fourier transformations of the velocity auto-correlation functions (VACFs) [45, 46]. The VDOSs, denoted as $g(\omega)$, as well as those normalised by ω^2 , represented as $g(\omega)/\omega^2$, are shown in Supplementary Fig. 12. It is evident that three amorphous states exhibit distinct VDOSs. In the case of VHDA, there is a prominent quasi-elastic component in $g(\omega)/\omega^2$, indicating its instability. This feature of VHDA aligns with its disordered local structures, which make it susceptible to crystallisation. Even in the case of HDA, a similar quasi-elastic component in $g(\omega)/\omega^2$ is observed shortly after the transformation from LDA, but it gradually diminishes as mechanical stability is acquired over time (see Supplementary Fig. 13). In the future, it will be essential to conduct a comprehensive study on the thermodynamic and mechanical stability of these high-pressure amorphous forms, especially VHDA.”

Page 6, middle: “This suggests that the locally favoured structure of HDA has octahedral symmetry”. How should one interpret this in terms of the clear presence of 60° angles in the bond angle distribution (BAD) in Fig. 1c? These would not be expected in the normal “perfect” octahedral case.

Reply:

We believe that the locally favoured structure of HDA possesses octahedral symmetry, and we do not consider it problematic. The reasons supporting our view are outlined below:

The coordination number (CN) in amorphous solids is widely acknowledged to exhibit a spectrum of values as a result of disorder effects, in contrast to the distinct single values observed in crystals. For example, the locally favoured motif in LDA of silicon is commonly recognized as a tetrahedron characterized by an approximate bond angle of 109 degrees. In practice, however, in the case of LDA, alongside the prominent peak, the BADF reveals a minor peak at approximately 55 degrees. This minor peak can be attributed to atoms with a coordination number (CN) of 5. Similarly, in the context of HDA, the locally favoured structure is expected to take on an octahedral configuration. This is evidenced by the prominent peak at approximately 90 degrees in the BADF (Fig. 1c) and corroborated by the findings presented in previous Supplementary Fig. 7b or current Supplementary Fig. 9b.

The minor peak observed at approximately 60 degrees in the BADF (Fig. 1c) can be **partly** attributed to atoms with $CN \neq 6$ in HDA. The CN distribution can be seen in Fig. 1d. In addition, we noticed that there is also a small peak around 60 degrees in the BADF exclusively for atoms with $CN = 6$ in HDA (see current Supplementary Fig. 8b or previous Supplementary Fig. 20b). However, this is neither contradictory to the fact that the locally favoured structure of HDA possesses octahedral symmetry, as the local motif in glasses are more or less distorted. In Fig. R11, we showcase characteristic octahedral environments with varying degrees of distortion surrounding atoms possessing a coordination number (CN) of 6 in the HDA form. The highly distorted octahedral environments may contribute a small peak around 60 degrees in the BADF for atoms with $CN = 6$ in the HDA form.

This information has also been incorporated into our revised manuscript to enhance clarity and comprehension as follows (see in the bottom of page 6).

“Supplementary Fig. 7 illustrates characteristic octahedral environments displaying different levels of distortion surrounding atoms that exhibit a CN of 6 in the HDA form.

The small peak observed around 60° in the BADF can be attributed to atoms with $CN \neq 6$, and it signifies distortions occurring within octahedral structures. These distortions may be closely linked to the structural instability that leads to higher-coordination structures or configurations resembling VHDA structures (Supplementary Fig. 8b).”

We hope the above explanations have effectively addressed the concerns raised by the reviewer.

Fig. R11 Typical octahedral environments with different RMSD relative to a perfect octahedron surrounding atoms with CN = 6 in the HDA form of α -Si. Here, the pairs of atoms with an interatomic distance $\leq 3.72 \text{ \AA}$ are connected with a bond. See Supplementary Note 5 for the definition of RMSD.

Page 8: “This provides more support to a positive answer...”. While I think I understand the meaning of the sentence I encourage rephrasing and elaborating the sentence to make it more clear to the reader.

Reply:

We have expanded upon the explanations regarding the change in the order parameter during the AAT (LDA-to-HDA transition) and its relevance as a similarity to thermodynamic transitions.

In light of these additional details, we have revised this paragraph accordingly, please see below and page 9.

“We find that the structural changes occurring during the LDA-HDA transition can be effectively characterized by a systematic temporal evolution of the distribution of the local bond orientational order parameter \bar{q}_4^{21} . Specifically, a new peak emerges around $\bar{q}_4^{21} =$

0.02 and steadily increases while maintaining its position (as shown in Fig. 2e). This behaviour is reminiscent of the order parameter evolution in the thermodynamic NG process [8,22,47]. Therefore, \bar{q}_4^{21} can serve as a suitable local structural order parameter [8,22] to effectively describe the LDA-HDA transition. This discovery of the NG-like features of AAT further supports the notion that AAT shares similarities with genuine thermodynamic phase transitions, which has been recognized as an intriguing and fundamental question in materials science [21]. However, it is important to emphasize that the AAT process in a solid state also involves mechanical contributions, as we will demonstrate later on.”

Page 8: “The 151 atoms which would form the HDA nucleus upon pressurization are indeed the atoms with relatively lower q_4^{21} in the initial sample”. This is a very interesting observation and I think the authors delve too little into this and related connections between initial structure and subsequent structural change upon pressurization. See my following points.

Although the authors do a somewhat extensive job describing the structural response under the various pressurization cases I suggest them to look deeper into the structural/dynamical origin of the points of nucleation. I imagine several ways to target this: 1) They could compute the eigenvectors using the dynamical matrix and from that gain fundamental knowledge of the vibrations of the atoms which later initiate phase change: Are the atoms vibrationally localized? Do nearby atoms move in or out of phase? 2) Given the authors interest in the use of machine learning the authors may look into the structural descriptor of so-called “softness” (see <https://doi.org/10.1038/nphys3644> and <https://doi.org/10.1103/PhysRevLett.122.028001>). Softness has so far seen very interesting usage for example to predict the nucleation sites in a simulated colloidal glass (<https://doi.org/10.1038/s41567-020-1016-4>). I wonder if softness could be able to predict the structural environments, the initiation sites for structural transitions, and perhaps even as a unified metric for the separation between LDA, HDA, and VHDA phases.

Reply:

We thank the reviewer for the thoughtful comments and suggestions. We fully agree that this phenomenon is very interesting. As discussed in Ref. [russo2016crystal], the role of preordering in triggering crystallisation in a supercooled liquid should be attributed to its thermodynamic origin rather than its mechanical counterpart. However, in the present case, the solid-state transformation is triggered by applying high pressure. Thus, as the reviewer correctly pointed out, softness can also be an important factor in initiating AAT. Therefore, whether the HDA-like preorder initiates the nucleation of HDA domains thermodynamically and mechanically is an intriguing question.

The analyses using vibrational modes and softness exhibit a higher sensitivity towards mechanically weak parts, which is evident from their effectiveness as predictors of plastic deformation. However, since the HDA-like order is more stable at high pressure, it is

evident without any further analysis that it should be favoured under compression. We are also sure that the local orientational order and softness both tend to trigger the nucleation of HDA. The concept of softness proves highly effective for systems governed by isotropic potentials, such as colloidal systems, as demonstrated in the references provided by the reviewer. However, its suitability for systems characterized by directional bonding remains uncertain. Additionally, the computational demands associated with diagonalizing the dynamic matrix are considerably high. The task of disentangling thermodynamic and mechanical effects is also expected to be challenging. In view of these considerations, we have only included a discussion on this matter in our revised manuscript while leaving its resolution to future research.

We have added the following sentences to address this intriguing point (page 10):

“The significant role played by preordering in initiating AAT shares similarities with the influence of crystal-like preordering in crystal nucleation [24,25,42]. This underscores the general significance of preordering in first-order phase transitions. While thermodynamic ordering from a liquid is often driven by preordering, with a primary thermodynamic impact, the situation can be substantially different in pressure-induced solid-state phase transitions due to their mechanical nature. In this context, understanding how HDA-like preordering catalyzes the nucleation of HDA domains, both thermodynamically and mechanically, presents an intriguing question. In addition to local symmetry matching, which reduces the energy barrier associated with the formation of new interfaces, the lower mechanical stability of preordered regions plays a crucial role in reducing the mechanical work involved in the transformation. Given the enhanced stability of HDA-like order under high pressure, it is intuitively expected that the transformation of HDA-like preordering into HDA domains becomes more feasible under compression. The intricate interplay between thermodynamic and mechanical factors within this transformation poses an intriguing question. However, due to the complexity of disentangling these intricate contributions, we leave this issue for future investigations.”

As the authors describe a lot of different pressurization and heating cases with varying structural output, it remains particularly hard to keep track of all these cases solely from the text. While the authors do briefly discuss some of my thoughts in Supplementary note 3 and Supplementary Figure 14, perhaps they could construct a type of “phase diagram” as a (supplementary?) figure to provide a general overview of which phases are present and stable under which conditions (P, rate of P, T).

Reply:

Following the reviewer’s suggestion, we have included a summary diagram as Fig. 7 in our revised manuscript, which is also shown below as Fig. R12.

Fig. R12 A schematic diagram illustrating the structural transformation paths among the three amorphous forms of a-Si and their crystallization paths investigated in the current work. The typical local structures of the three amorphous forms and two crystals of silicon are also shown.

Supplementary Figure 20a: Both the HDA and VHDA phases are seen to feature very abrupt changes in $g(r)$ (e.g., VHDA CN=6 and HDA CN=7). Do the authors believe this is physical or an artefact of the simulation/potential?

Reply:

It is a common occurrence to observe abrupt changes in the radial distribution function ($g(r)$) when calculating $g(r)$ exclusively for atoms with a coordination number (CN) equal to a specific value. However, it is important to note that there is no such discontinuity present in the $g(r)$ profile when considering all atoms, as depicted in Figure 1b.

Supplementary Figure 20b: It is interesting to observe that in the VHDA phase, the lowest bond angles ($\sim 60^\circ$) are higher than in the HDA case. Perhaps the authors could dig into this effect by accessing the per atom stress in the two phases?

Reply:

The difference in the Bond Angle Distribution Function (BADF) between HDA and VHDA for atoms sharing the same coordination number (CN) signifies a difference in the extent of distortion of local polyhedra between these two amorphous states. This disparity is also observable in the discernible dissimilarity in middle-range structural order, which is evident through the radial distribution function ($g(r)$) displayed in Figure 1b and the ring analysis provided in previous Supplementary Figure 17 or current Supplementary Figure 10. We agree that unveiling a mechanical origin through the examination of atomic-level stress would be intriguing; however, the precise evaluation of atomic-level stress causing a subtle structural difference under thermal noise is challenging. Thus we have opted to leave this particular question for future investigations.

We hope that we have effectively addressed the reviewer's comments and that the revised manuscript is now well-suited for publication in Nature Communications.

REVIEWERS' COMMENTS

Reviewer #1 (Remarks to the Author):

The authors addressed all my questions, and provide a detailed reply to the points raised by all referees. This work provide insights on the amorphous-amorphous phase transition in silicon that can be extended to another materials and substances with this kind of phase transition. Then, the manuscript can be accepted as it is. However, a transfer to Physics Communications, as suggested by Reviewer #4, is applicable if the editor deems it appropriate.

Reviewer #2 (Remarks to the Author):

The authors have addressed my questions and comments. I am happy to support publication of the manuscript in Nature Communications.

Reviewer #3 (Remarks to the Author):

The authors replied to my questions/comments and revised the manuscript in a satisfactory manner. As I wrote in the previous report, I appreciate the work in terms of new results elucidated by non-equilibrium protocols. But as the authors also admitted, there is no novelty in terms of methodological/technical aspects.

Below, I will summarize my precious concerns and conclude my statement.

- 1) Distinctions between nucleation-growth (NG) and spinodal decomposition (SD) type mechanisms were unclear in the previous submission. Now, the authors have explained this issue carefully.
- 2) The comparison between a liquid-liquid transition seen in an experiment and an amorphous-amorphous transition in this manuscript does not make sense since the associated lengthscales are very different. This paragraph was removed by the authors in the current submission.
- 3) I had a concern about the wording "mechanical" because they did not study any standard mechanical features in the previous manuscript. The authors now explained carefully what "mechanical" means in the context of this manuscript.

Besides, it seems that the authors responded to most of the questions and comments from the other referees carefully and properly. Reviewer 4 asked the authors to perform additional analysis in terms of softness measured in a machine learning method (support vector machine). However, finding a local structure associated with the initiation of macroscopic transformation would be out of the scope of this current manuscript. So, what the authors responded seems reasonable to me.

To conclude, I now recommend the manuscript for publication in Nat. Commun.

Reviewer #4 (Remarks to the Author):

I wish to thank the authors for properly addressing the questions and concerns of my initial review.

I believe their response and changes are sound and significantly improves the manuscript. Therefore, I now recommend its publication.